# m$^5$C methylated *lncRncr3*–MeCP2 interaction restricts *miR124a*-initiated neurogenesis

Jing Zhang [1] ✉, Huili Li [1] & Lee A. Niswander [1] ✉

Coordination of neuronal differentiation with expansion of the neuroepithelial/neural progenitor cell (NEPC/NPC) pool is essential in early brain development. Our in vitro and in vivo studies identify independent and opposing roles for two neural-specific and differentially expressed non-coding RNAs derived from the same locus: the evolutionarily conserved lncRNA *Rncr3* and the embedded microRNA *miR124a-1*. *Rncr3* regulates NEPC/NPC proliferation and controls the biogenesis of *miR124a*, which determines neuronal differentiation. *Rncr3* conserved exons 2/3 are cytosine methylated and bound by methyl-CpG binding protein MeCP2, which restricts expression of *miR124a* embedded in exon 4 to prevent premature neuronal differentiation, and to orchestrate proper brain growth. MeCP2 directly binds cytosine-methylated *Rncr3* through previously unrecognized lysine residues and suppresses *miR124a* processing by recruiting PTBP1 to block access of DROSHA-DGCR8. Thus, miRNA processing is controlled by lncRNA m$^5$C methylation along with the defined m$^5$C epitranscriptomic RNA reader protein MeCP2 to coordinate brain development.

Cerebral cortical development starts with highly proliferative embryonic neuroepithelial progenitor cells (NEPC) and subsequent neural progenitor cells (NPC)/radial glial cells, followed by progressive differentiation into neurons and glia[1,2]. Expansion of the NEPC and NPC pools and temporal control of neuronal differentiation is critical to ensure an appropriate population of neural stem cells to produce sufficient neurons and glia at later stages[2–6]. Either premature commitment to neurogenic division or death of NPCs can deplete the neural progenitor pool and thus cause reduced brain size (microcephaly)[2,5]. Notch signaling maintains NPC character and inhibits neuronal differentiation in fruit fly and mice[1,7]. Cell fate determinants NUMB/BRAT turn off Notch signaling and promote neuronal differentiation in fruit fly[1]. In parallel to coding genes, accumulating evidence shows epigenetic, epitranscriptomic, and alternative splicing can temporally control cortical neurogenesis[8–10]. In vivo deletion of long non-coding RNA (lncRNA) *Evf2*, *Pnky* or *linc-Brn1b* leads to disordered cortical development[11–13].

Long non-coding RNAs represent the vast majority of cell-specific transcription[14,15] and have a wide range of activities. Mutations in lncRNAs in mice show visible phenotypes, for example in brain development, limb development, and the immune system, and can affect viability and genome stability[16]. lncRNAs can function within the nucleus or the cytoplasm to recruit various proteins such as transcription factors, chromatin remodelers, polymerase, and proteins that introduce post-translational modifications. Hence lncRNAs often act indirectly to affect chromatin organization, transcription, mRNA stability, and protein activity[14–16]. The epitranscriptome refers to dynamic biochemical modifications of RNA. Previous studies have implicated the temporal control of mammalian cortical neurogenesis through mRNA m$^6$A methylation or mitochondrial tRNA m$^3$C modification[17–19]. 5-methylcytosine (also called 5-methylcytidine) is well studied in DNA (5mC) and can occur on RNA (m$^5$C) at levels ranging from 0.03-0.1% of cytosines[20]. The m$^5$C epitranscriptome influences development and cancer[21,22]. Two m$^5$C methyltransferases NSUN2 and TRDMT1/DNMT2 catalyze m$^5$C modification on specific tRNAs, mRNAs and lncRNAs[23–25]. Moreover, NSUN2 mediates m$^5$C methylation of vault non-coding RNAs and of *pri-miR125b* to control their processing into regulatory small RNAs[23]. Loss of NSUN2 results in

[1]Department of Molecular, Cellular, and Developmental Biology. University of Colorado Boulder, Boulder, CO 80309, USA.
✉e-mail: jing.i.zhang@colorado.edu; kinnzhang@gmail.com; lee.niswander@colorado.edu

reduced thickness and disorganization of the cerebral cortex[26,27], yet there is little information on the lncRNAs that are modified by m5C or how the m5C modification may change their functionality in the developing brain.

LncRNAs can also serve as host transcripts for microRNAs. MicroRNA *miR124a* is the most abundant miRNA in the developing brain[28–30]. *miR124* can initiate neuronal differentiation and determine cell fate during neurogenesis and in the postnatal brain[28,31,32]. *miR-124* in combination with *miR-9* can convert human fibroblasts into physiologically functional neurons[32,33]. *miR124a-1* is embedded within exon 4 of *lncRncr3* (retinal non-coding RNA3; hereafter called *Rncr3* and *miR124a*). One mechanism of restricting *miR-124a* processing is the binding of the polypyrimidine tract-binding protein PTBP1 to a CU-rich region upstream of the miRNA stem-loop in *Rncr3* exon 4 to inhibit binding of the DROSHA/DGCR8 nuclear miRNA processing complex[34]. Yet, in the developing brain, *Rncr3* and *miR124* do not always have overlapping patterns of expression[35]. Deletion of the entire *Rncr3-miR124a* locus in mice results in postnatal reduction in brain size, limb clasping, apoptosis of cells in the dentate gyrus and cortex, and retinal defects, and these phenotypes were attributed to loss of *miR124a* function[35]. Yet only some deficits were rescued by *miR124a* reintroduction, whereas the reduction in brain size, limb clasping, and cortical apoptosis were not rescued[35]. Thus, we asked whether *Rncr3* acts independently of *miR124a* in brain development.

Mutations in the methyl-CpG binding protein 2 (MeCP2) are the major cause of Rett syndrome, a neurological disorder that includes reduction in brain volume, gait abnormalities and stereotypic movements[36–42]. Similar phenotypes are seen upon loss of the entire *Rncr3* locus in mice[35]. Although MeCP2 is best known for binding CpG methylated DNA and influencing the expression of an adjacent gene, MeCP2 also binds coding and non-coding RNAs in the brain, including with high affinity to *Rncr3*[43], although the mechanism underlying the physical association of RNAs and MeCP2 remains poorly understood[44,45]. Furthermore, MeCP2 is recognized as contributing to the regulation of miRNA production: phosphorylated MeCP2 can bind to DGCR8 and interfere with the assembly of the DROSHA/DGCR8 complex to inhibit processing of primary transcripts to precursor miRNAs[46,47]. Yet, whether this mechanism of MeCP2-DGCR8 interaction underlies all cases of MeCP2 negative regulation of miRNA processing is unclear. Moreover, MeCP2 has also been associated with positive regulation of miRNA processing[48], although the mechanistic underpinning of this regulation remains unknown. The broader roles for MeCP2 beyond its traditional role in binding cytosine methylated DNA, highlights the need to more fully understand the impact of MeCP2 in the brain, in particular as it relates to RNA function. These pieces of evidence led us to explore the functional relationships among MeCP2, *Rncr3* and *miR124a*.

Through molecular and genetic manipulation, our studies identify independent and opposing roles for these two neural-specific and differentially expressed non-coding RNAs derived from the same locus: *Rncr3* is expressed in and necessary for embryonic NEPC/NPC proliferation and survival whereas *miR124a* expression initiates at the onset of and promotes neuronal differentiation. Expression of *miR124a* embedded in exon 4 is limited by *Rncr3* conserved exons 2/3. In vivo deletion of *Rncr3* exons 2/3 (leaving *miR124a* intact) in heterozygous mice results in a spectrum of defects including reduced brain weight, premature neuronal differentiation, decreased NEPC/NPC proliferation, and increased NEPC/NPC death, altogether reducing the neural progenitor pools. Mechanistically, *Rncr3* exons 2/3 undergo m5C modification by RNA methyltransferases and this is essential for MeCP2 binding. Moreover, the MeCP2 methyl binding domain is not required and instead we identify a stretch of lysines in the intervening domain of MeCP2 needed for m5C *Rncr3* binding. Direct binding of MeCP2 to *Rncr3* exons 2/3 recruits PTBP1 to suppress *miR124a* processing. Taken together, our studies in both mouse and human cells highlight that these non-coding RNAs from the same locus have opposing functions in NEPC/NPC proliferation and differentiation, that the m5C modified *Rncr3* exons 2/3 are critical in the suppression of *miR124* to avoid premature neuronal differentiation, and that MeCP2 is a cytosine methylated RNA reader protein with significance for clinical assessment as well as RNA biology.

## Results

### Rncr3 exons 2/3 drive NEPC/NPC proliferation and survival independently of miR124a

LncRNAs usually display little nucleotide conservation across species[49]. The human ortholog *MIR124-1* is 100% conserved over the pre-miR stem-loop (85 nucleotides), and is embedded in exon 5 of lncRNA *LINC00599* (ENST00000685227.2)[34,35]. Although the sequence of *miR-124* is conserved across fish to human, conservation of the locus is limited to mammals (Fig. 1a), and in particular exons 2 and 3 are highly conserved between mouse *lncRncr3* and human *LINC00599* (hereafter called *RNCR3*) (76% and 83% identity over >100 nucleotides, respectively, Fig. 1b and Supplementary Fig. 1a). Some lncRNAs have the potential to encode small peptides, but analysis of *Rncr3* or conserved exons 2 and 3 does not support coding probability (Supplementary Fig. 1b using coding potential assessment tool (CPAT) and GWIPS-viz ribosome footprinting RNA-seq database)[50].

In mouse embryos, *Rncr3* (exons 2/3 probed) is expressed in undifferentiated NEPCs at E9.5–E10.5, whereas *miR124a* is not detected until the onset of neuronal differentiation (E11.5–E12.5; Fig. 1c). We first confirmed the role for *miR124a* in neuronal differentiation using NE-4C cells, a NEPC cell line from cerebral vesicles of embryonic day 9 (E9) mouse embryos. *miR124a* inhibitor strongly suppressed, and *miR124a* mimic promoted, neuronal differentiation, however, neither treatment impacted NEPC proliferation or survival (Supplementary Fig. 1c–f). To evaluate whether *Rncr3* functions in NEPCs, anti-sense oligonucleotide (ASO) knock-down (KD) was used to decrease *Rncr3/RNCR3* in proliferating NE-4C or in ReNcell CX cells, a human fetal NPC cell line from the cortical region (Supplementary Fig. 1g, h). We also used CRISPR/Cas9 gene editing of NE-4C cells to create full-length deletion of the *Rncr3−miR124a* locus (exons 1–4) (Fig. 1b; Supplementary Fig. 2a). Decreased expression or loss of the entire *Rncr3* locus significantly attenuated NEPC and NPC proliferation (pH3 and BrdU) and enhanced apoptosis (cleaved Caspase-3 or TUNEL) compared to controls (Fig. 1d and Supplementary Fig. 2b–c).

To specifically address whether conserved exons 2 and 3 are functionally important in neural development, we deleted these exons in NE-4C cells (leaving *miR124a* sequence intact; Fig. 1b; Supplementary Fig. 2d). Exons 2/3 deletion notably reduced NEPC proliferation and survival (Fig. 1e). To comprehensively evaluate the effects of full-length *Rncr3−miR124a* KO and exons 2/3 deletion, we performed RNA-seq on the NE-4C deletion cell lines under proliferation conditions. We first focused on genes whose expression was consistently up- or down-regulated in full-length KO and exons 2/3 deletion clones relative to wildtype (Supplementary Fig. 2e). Gene ontology (GO) Biological Process (BP) functional annotation showed enrichment for negative regulation of cell proliferation; cell fate commitment; cell cycle arrest; negative regulation of apoptotic process (Supplementary Fig. 2f; gene heat map comparisons in Supplementary Fig. 2g). The RNA-seq data supports the observed cellular functions of *Rncr3* and indicate *Rncr3* functions in NEPC/NPC pool expansion but that *miR124a* is not a pivotal component.

To translate these findings in vivo, we first used ASO KD and *ex utero* cultures of E8.5 mouse embryos. As above, this disrupted NEPC proliferation and survival (Supplementary Fig. 2h, i). We then generated a mouse strain with specific deletion of exons 2/3. Embryos homozygous for the deletion died before organogenesis (E8.5), while heterozygous embryos showed reduced body size, reduced brain weight, and abnormal body posture at E14.5 and 18.5 (Fig. 1f, g). At

E12.5, when NPCs are still actively proliferating but neuronal differentiation has begun, there was a significant decrease in the NEPC/NPC marker PAX6 and proliferation (pH3) in heterozygous exons 2/3

deletion NPCs in the ventricular zone (VZ), and an increase of cell death (cleaved-Caspase3) throughout the cortex (Fig. 1h–j and Supplementary Fig. 2j). Together our cell line studies, RNA-seq data, and

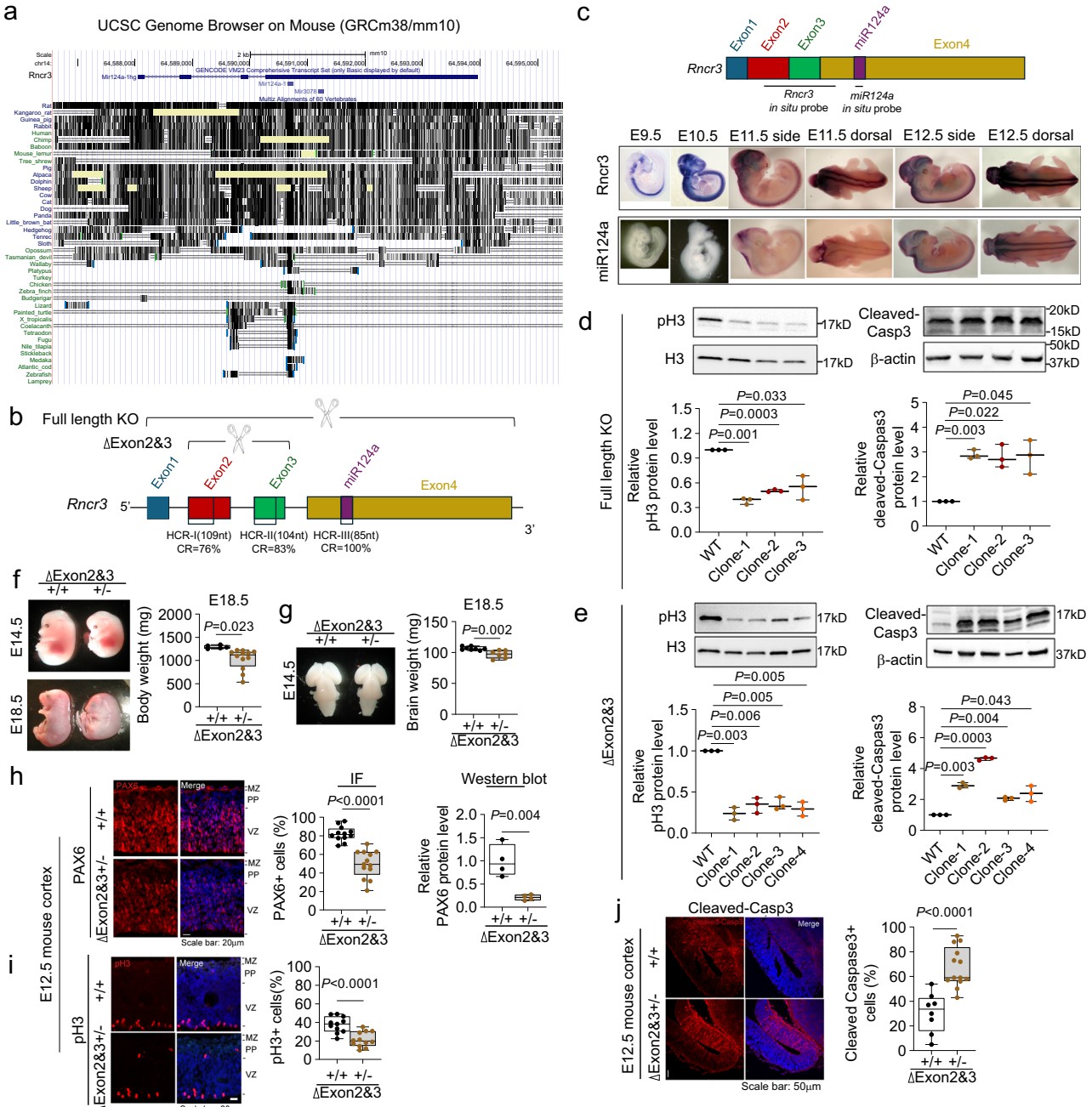

**Fig. 1 | Conserved *Rncr3* exons 2/3 independently maintain NEPCs and NPCs.** **a** UCSC Genome Browser on Mouse (GRCm38/mm10) and alignment of sequences around *miR142a-1* across mammalian and non-mammalian species. The sequence of *miR-124* is conserved from fish to human, however the rest of the gene body (*MiR124a-1hg*/*Rncr3*) is conserved only in mammals. **b** Highly conserved regions (HCR) in *Rncr3* across human and mouse: exon 4 around *miR124a* (85 nt, 100% conservation), exon 2 (191 nt) and exon 3 (120 nt) are 64% and 78% conserved and show higher conservation over 109 nt in exon 2 (76% conservation) and 104 nt in exon 3 (83% conservation): Top illustrates the *Rncr3* exons 2/3 deletion or *Rncr3-miR124a* knockout (exons 1-4). **c** Whole mount in situ hybridizations for *Rncr3* and mature *miR124a* expression in mouse embryos from embryonic day 9.5–12.5. Representative immunoblots and quantification of proliferation marker pH3 and apoptosis marker cleaved-Caspase3 in *Rncr3* full length knockout, KO (**d**) or exons 2/3 deletion (**e**) NE-4C cell clones (*n* = 3 independent experiments for each clone)

under proliferation conditions. **f, g,** Heterozygous (+/-) mouse embryos carrying a deletion of *Rncr3* exons 2/3 show reduced body and brain weight and abnormal posture (**f**, wildtype, +/+ *n* = 5 embryos, heterozygous +/- *n* = 12; **g**, +/+ *n* = 8, +/- *n* = 7). **h, i, j** Decreased NPC pool in cerebral cortex of *Rncr3* exons 2/3 deletion heterozygous embryos. Confocal images of coronal sections from E12.5 wild-type and exons 2/3 deletion heterozygous cortex stained with antibodies against PAX6 (**h**, red; +/+ n = 12, +/- n = 13, scale bar: 20 μm), pH3 (**i**, red; +/+ n = 11, +/- n = 12, scale bar: 20 μm), and cleaved-Caspase3 (**j**, red; +/+ n = 8, +/- n = 13, scale bar: 50 μm) with quantification of immunofluorescent staining. Western blot for PAX6 protein see Supplementary Fig. 2j, *n* = 4 embryos. Hoechst stains nuclei (blue). **d–j** Student's *t* test, two-sided; boxplots show all data points with box which extends from the 25th to 75th percentiles and whiskers showing min to max, and the line in the middle of the box is plotted at the median. Source data are provided as a Source Data file.

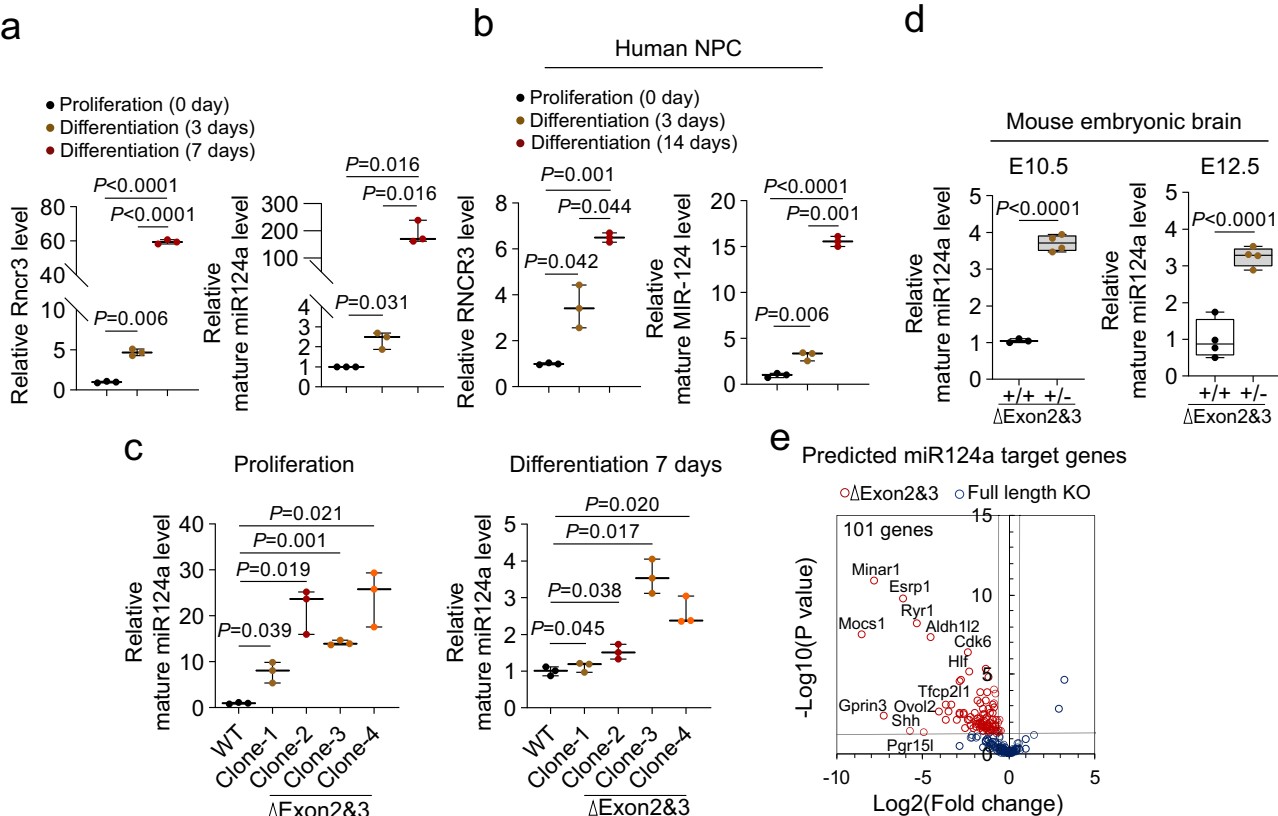

**Fig. 2 | *Rncr3* exons 2/3 regulate *miR124a* production. a** Quantitative RT-PCR for *Rncr3* and *miR124a* expression levels in NE-4C cells under proliferation conditions and 3 and 7 days after induction of neuronal differentiation (*n* = 3 biologically independent samples). **b** Quantitative RT-PCR of human ReNcell CX cells for *RNCR3* and *MIR124-1* expression levels under proliferation conditions and 3 and 14 days after induction of neuronal differentiation (*n* = 3 biologically independent samples). **c** Quantitative RT-PCR shows the level of mature *miR124a* in wildtype and *Rncr3* exons 2/3 deletion NE-4C clones under proliferation or differentiation conditions (*n* = 3 biologically independent samples). **d** Quantitative RT-PCR shows increased expression of mature *miR124a* in *Rncr3* exons 2/3 deletion heterozygous

(+/-) versus wildtype (+/+) mouse embryo brains at E10.5 and E12.5 (E10.5, +/+ *n* = 3, +/- *n* = 4 embryos, E12.5, +/+ *n* = 4, +/- *n* = 4,); For **a**–**d**, Student's *t* test, two-sided; boxplots show all data points with box which extends from the 25th to 75th percentiles and whiskers showing min to max, and the line in the middle of the box is plotted at the median. **e** Predicted *miR124a* target genes aligned with RNA-seq data and shown as a scatter plot of log2 (fold change) and -log10 (adjusted *P* value, also known as q-value produced from multiple testing correction of False Discovery Rate, FDR). A total of 101 target genes are downregulated in *Rncr3* exons 2/3 deletion (red circles) but not changed in full length deletion NE-4C cells (blue circles). Source data are provided as a Source Data file.

in vivo mouse data support the importance of exons 2/3 in maintenance of NEPCs/NPCs.

## *Rncr3* exons 2/3 regulate *miR124a* production

The difference in expression of the host transcript *Rncr3* in mouse embryos and in mouse and human NEPCs/NPCs versus *miR124a/ MIR124-1* expression during neuronal differentiation (Fig. 1c, Fig. 2a, b) prompted exploration of the regulation of *miR124a* production. In exons 2/3 deletion cells under proliferation conditions, there is a sharp increase and premature expression of mature *miR124a* levels (8–24 fold; Fig. 2c) that is not explained by increased transcription of the locus (exon 1 and *miR124a* region in exon 4: 1–3 fold relative to control cells) (Supplementary Fig. 3a, see Methods). Likewise, in exons 2/3 deletion heterozygous mouse embryo brains, we observed increased *miR124a* expression at E10.5 and E12.5 (Fig. 2d). These data suggest that *Rncr3* exons 2/3 regulate expression of *miR124a*.

As the *miR124a* sequence is retained in exons 2/3 deletion cells and *miR124a* drives differentiation, we questioned whether some downregulated genes in exons 2/3 deletion cells may be *miR124a* target genes. *TargetScanMouse* website predicted 117 downregulated genes were *miR124a* targets (Fig. 2e, see methods), of which a few highlights are *Minar1* which interacts with NOTCH2 to increase its stability and function[51]; and *Cdk6* a cyclin-dependent kinase involved

in G1 phase progression and G1/S transition[5,52,53]. In full-length KO cells, the vast majority of these predicted *miR124a* targets (101 genes) showed no significant change (Fig. 2e). Together our data show that (1) even though *lncRncr3* is the host transcript for *miR124a*, there are temporal differences in their expression during early neural development, (2) deletion of exons 2/3 leads to premature *miR124a* expression, and suggest (3) exons 2/3 negatively regulate *miR124a* expression.

## *Rncr3* exons 2/3 regulate neuronal differentiation and embryonic corticogenesis

Further examination of the RNA-seq data showed that exons 2/3 deletion NE-4C cells, but not full-length KO cells, had significant enrichment for genes associated with neuronal differentiation, despite being grown under proliferation conditions (GO enrichment analysis; Fig. 3a, b and Supplementary Fig. 4a), consistent with the premature induction of *miR124a*. A time course over proliferation and differentiation conditions showed premature and increased expression of neuronal markers in exons 2/3 deletion versus wildtype cells, whereas full-length KO cells showed minimal induction (Fig. 3c, d and Supplementary Fig. 4b, c), and inhibition of *miR124a* rescued the premature neuronal differentiation phenotype (Supplementary Fig. 4d). In vivo we observed premature expression of neuronal differentiation markers (NFM and TUJ1), and decreased levels of PAX6 and pH3

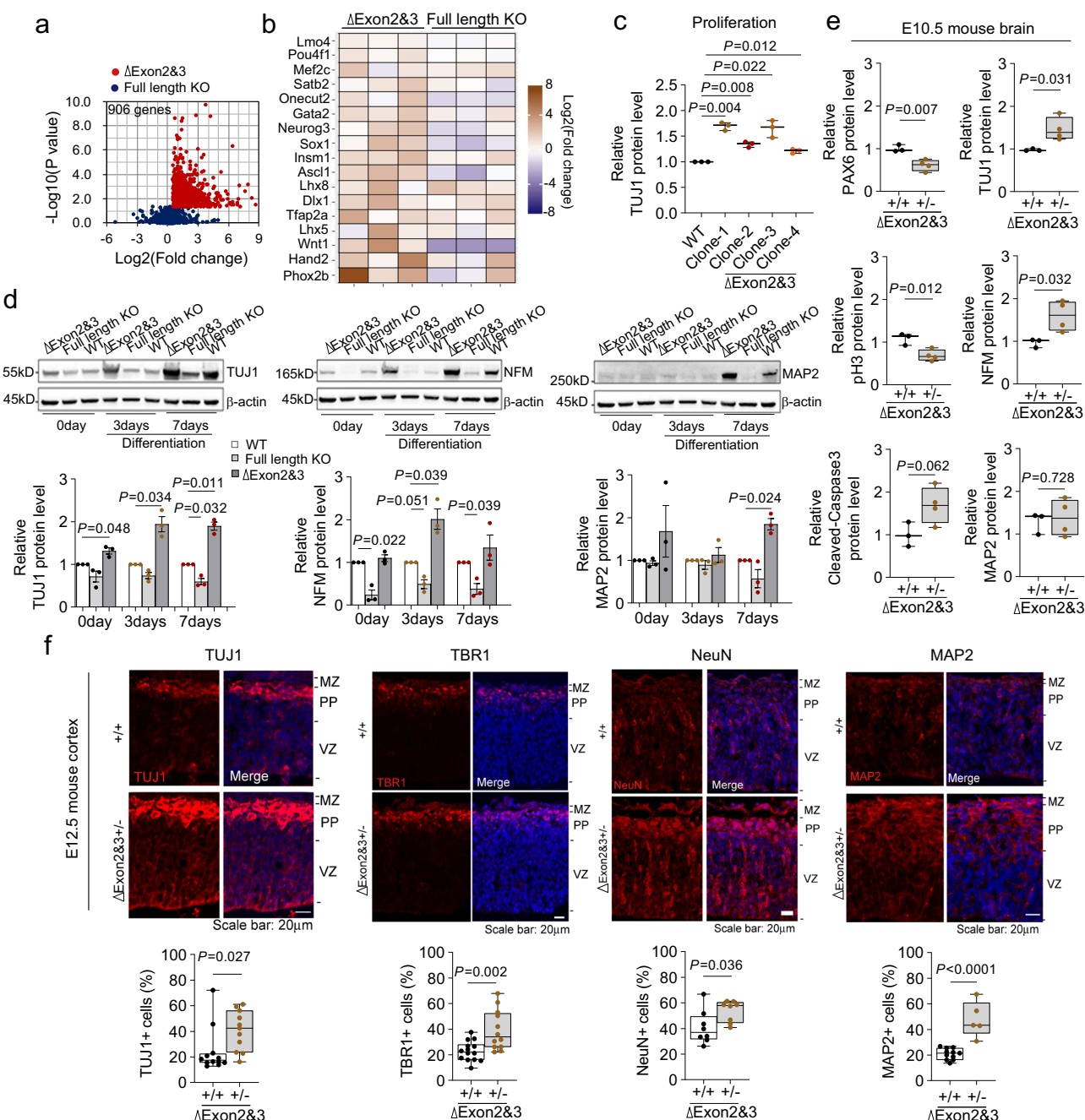

**Fig. 3 | *Rncr3* exons 2/3 regulate neuronal differentiation and embryonic corticogenesis. a** Scatter plot of RNA-seq data (as defined in Fig. 2e) showing total 906 genes upregulated in exons 2/3 deletion NE-4C cells (red dots) but not changed in full length deletion (blue dots). **b** Heat map of genes enriched in neuronal differentiation (three independent clones for each genotype). **c** Quantification of immunoblot data in Supplementary Fig. 4b shows increased protein level of neuronal marker TUJ1 in *Rncr3* exons 2/3 deletion NE-4C cells under proliferation conditions (*n* = 3 independent experiments for each clone). **d** Immunoblot data and quantification of TUJ1, NFM and MAP2 expression levels under a time course of neuronal differentiation in *Rncr3* exons 2/3 deletion or full length deletion clones compared to wildtype NE-4C cells. Student's *t* test, two-sided; mean ± s.e.m. *n* = 3 biologically independent experiments. **e** Quantification of immunoblots from

representative of NEPCs in E10.5 Δexons 2/3 heterozygous embryonic brains (Fig. 3e and Supplementary Fig. 4e). In E12.5 heterozygous cortical sections, there were increased numbers of cells expressing neuronal differentiation markers TUJ1, TBR1, NeuN, and MAP2 and

Supplementary Fig. 4e of NEPC/NPC markers PAX6 and pH3, apoptosis marker (cleaved-Caspase3) and neuronal differentiation markers TUJ1, NFM and MAP2 in *Rncr3* exons 2/3 deletion heterozygous (+/-) mouse embryo brains (E10.5) *versus* wildtype (+/+). + /+ *n* = 3, +/- *n* = 4 embryos. **f** Coronal sections from E12.5 wildtype (+/+) and exons 2/3 deletion heterozygous (+/-) cortex stained with antibodies against TUJ1, TBR1, NeuN and MAP2, and cell numbers quantified below. Hoechst stains nuclei (blue). For TUJ1, +/+ n = 12, +/- *n* = 11; for TBR1, +/+ n = 14, +/- *n* = 12; for NeuN, +/+ n = 8, +/- *n* = 8; for MAP2, +/+ n = 11, +/- *n* = 5. Scale bar: 20 μm. In (**c**, **e**, **f**), Student's *t* test, two-sided; boxplots show all data points with box which extends from the 25th to 75th percentiles and whiskers showing min to max, and the line in the middle of the box is plotted at the median. Source data are provided as a Source Data file.

NFM by Western blot (Fig. 3f and Supplementary Fig. 4f). These data suggest that restriction of *miR124a* processing by upstream exons 2/3 is essential in NEPCs/NPCs to coordinate neuronal differentiation with neural progenitor pool expansion.

## MeCP2 protein binds to cytosine methylated *Rncr3* exons 2/3 to regulate *miR124a* production

To explore the mechanism by which *Rncr3* exons 2/3 limit the expression of exon 4-embedded *miR124a*, we considered two proteins previously shown to interact with *Rncr3*. One is methyl-CpG binding protein 2 (MeCP2). *Rncr3* was captured in a screen for lncRNAs bound by MeCP2 although the mechanism and relevance of this interaction remains unknown[43]. MeCP2 is a risk gene for microcephaly and a range of neurodevelopmental disorders, including Rett syndrome, autism spectrum disorder, and neurodevelopmental regression[36,38,39,42,45]. Notably, a genome-wide study indicates *miR124* expression is upregulated in *Mecp2* mutant brain[46]. The second is polypyrimidine tract-binding protein PTBP1 which binds to the CU-rich region in exon 4 upstream of *pri-miR124a* to inhibit DROSHA assembly and cleavage and hence inhibits mature *miR124a* expression[34]. Thus, we questioned whether there is a regulatory circuit between these proteins and *Rncr3* exons 2/3 that controls *miR124a* expression and neuronal differentiation. Furthermore, in view of the wide role of MeCP2 binding to cytosine methylated DNA, we questioned whether MeCP2 can functionally bind to cytosine methylated RNA.

RNA immunoprecipitation (RIP) assays against MeCP2 protein show that MeCP2 preferentially binds to exons 2/3 compared to exon 4 (Fig. 4a). RIP enrichment for m5C methylation (m5C-RIP) shows *Rncr3* exons 2/3, but not exons 1 or 4 or *miR124a* region, are specifically cytosine methylated (Fig. 4b). In a dose-dependent manner, increasing levels of folic acid, which promotes cytosine methylation, increased m5C levels specifically within exons 2/3 (Supplementary Fig. 5a) and enriched for binding by MeCP2 (Supplementary Fig. 5b). Conversely, inhibition of cytosine methylation using 5-azacytidine (5AZA) or 3-deazaadenosine (3DAA) abolished pull-down of *Rncr3* exons 2/3 by m5C- or MeCP2-antibodies (Fig. 4c). Similarly, human NPC cells also show cytosine methylation-dependent MECP2 binding to RNCR3 conserved regions (Fig. 4d; Supplementary Fig. 5c). Knock-down of RNA m5C methyltransferases *Nsun2* or *Trdmt1*[23,24,54,55] decreased *Rncr3* m5C methylation levels and MeCP2 binding of *Rncr3* exons 2/3 (Fig. 4e and Supplementary Fig. 5d).

Using MeCP2-RIP bisulfite sequencing, we defined cytosine methylated sequences in *Rncr3* exons 2/3 and m5C sites preferentially bound by MeCP2. The data showed short C-rich sequences in exon 2 (135–175 with 14 cytosines) and in exon 3 (264–334 with 29 cytosines) can be methylated (Fig. 4f). Many of these cytosine residues are conserved in mammals and all cytosine residues at positions 271-313 within *Rncr3* exon 3 are conserved between mouse and human (Supplementary Fig. 5e). The specificity of the cytosine methylation pattern was confirmed by analyzing total RNA from proliferating NE-4C cells grown in the presence of 3DAA or siRNA KD of *Nsun2* or *Trdmt1* (Supplementary Fig. 5f). As cytosine residue 271 showed a high-frequency methylation pattern bound by MeCP2 (Supplementary Fig. 5g), we employed a simplified system using synthesized *Rncr3* RNA oligonucleotides with unmethylated or methylated cytosine-271, RNA pull-down and western blotting assays to directly show that methylation of cytosine 271 enhanced interaction with MeCP2 (Supplementary Fig. 6a). Moreover, we created MS2-tagged full-length *Rncr3* plasmids with C to T mutation of cytosine 271 or eight cytosines in region 271-313 and these mutations decreased interaction with MeCP2 (Supplementary Fig. 6b).

Notably, under neuronal differentiation conditions, the cytosine methylation modifications were not observed in 50 clones following bisulfite conversion of total NE-4C RNA (Supplementary Fig. 5g), corresponding with greatly decreased methylation of *Rncr3* exons 2/3 and significantly reduced physical interaction between exons 2/3 and MeCP2 (Fig. 4g). This change in methylation pattern upon differentiation was confirmed by Chromatin Isolation by RNA Purification (ChIRP) with western blotting for MeCP2 (Supplementary Fig. 6c). Of note, in mouse embryonic brains, m5C methylation of *Rncr3* exons 2/3

and MeCP2 binding significantly declined as neuronal differentiation initiated at E12.5 (Fig. 4h), this demonstrates that cytosine methylation of RNA promotes MeCP2 association.

This raised a new question as to whether cytosine methylation-dependent binding of *Rncr3* to MeCP2 is unique or a more general feature of MeCP2–RNA interaction? Combining a re-analysis of two published mouse brain datasets:(1) MeCP2 bound RNAs and (2) m5C modified mRNAs[43,56] shows a 70% overlap between these datasets (Supplementary Fig. 6d). Additionally, enrichment for MeCP2-bound RNAs from NE-4C cells followed by dot blot analysis for m5C showed significantly higher (~8-fold) m5C levels in MeCP2 protein-bound target RNAs versus input control (Supplementary Fig. 6e). These findings suggest that MeCP2 acts as a broad reader of the m5C epitranscriptome.

We next asked whether cytosine methylation of critical exons 2/3 and MeCP2 binding serves to regulate *miR124a* expression. Transfection of MS2-tagged C > T cytosine 271 or cytosines 271-313 plasmids showed increased mature *miR124a* levels compared to wildtype *Rncr3* plasmid (Supplementary Fig. 6f). Mature *miR124a* levels increased in NEPCs subjected to siRNA KD of *Mecp2* under proliferation conditions (Fig. 4i and Supplementary Fig. 6g). Similarly, inhibition of cytosine methylation by 3DAA or 5-AZA or siRNA KD against *Nsun2* or *Trdmt1* led to significant increases in *miR124a* expression (Fig. 4i and Supplementary Fig. 6h, i). Moreover, 5-AZA treatment of human NPCs increased *MIR124-1* levels (Fig. 4j). As controls, siRNA KD of *Mettl3* (N6-methyladenosine, m6A RNA methyltransferase) did not alter *miR124a* level (Supplementary Fig. 5d, 6j). Decreased methylation by folic acid insufficiency (0.026 mg/L) led to an increase in mature *miR124a* expression, even as overall levels of *Rncr3* decreased. Conversely, supraphysiological folic acid conditions led to increased *Rncr3* expression but mature *miR124a* was found at low levels (Supplementary Fig. 6k); which correlates with increased exons 2/3 methylation (Supplementary Fig. 5a). These data indicate that *Rncr3* exons 2/3 cytosine methylation is needed for interaction with MeCP2, and this MeCP2:*Rncr3* exons 2/3 interaction is necessary for suppression of *miR124a* processing in neural progenitors.

## MeCP2 recruits PTBP1 to restrict *miR124a* production

MeCP2 can regulate miRNA production through direct interaction with DGCR8 and interference with the assembly of DROSHA and DGCR8 nuclear microRNA-processing machinery complex[46]. However, our data indicates MeCP2 directly binds *Rncr3* to control *miR124a* processing, raising the possibility of a different mechanism whereby *Rncr3* exons 2/3 and MeCP2 may act in conjunction with PTBP1 to inhibit *miR124a* production in NEPCs/NPCs. PTBP1 binds a CU-rich region upstream of the miRNA stem-loop in *Rncr3* exon 4 to negatively regulate mature *miR124a* expression[34]. Compared with this 107 nucleotide CU-rich region in mouse *Rncr3*, a partially conserved CU-rich segment was identified in human *RNCR3* (79 nt, conserved identity 73%). Notably, a completely conserved PTBP1 binding motif CCUCUCU-CUCC is closely adjacent to the 5′ of *miR124a/MIR124-1* stem−loop in mouse and human pri-*miR-124-1* (Supplementary Fig. 7a). Using RIP, we found that PTBP1 binding is reduced in exons 2/3 deletion cells or by siRNA KD of *Mecp2* (Fig. 5a), and in human NPCs the binding between MECP2 and PTBP1 is increased by folic acid treatment and significantly decreased by 5AZA treatment (Fig. 5b). MeCP2 and PTBP1 physically interact as shown by co-IP and this interaction is largely dependent on exons 2/3 (Fig. 5c). Concurrently, there is increased access of DROSHA and DGCR8 to the *miR124a* cleavage site on exon 4 of *Rncr3* in exons 2/3 deletion cells or *Mecp2* KD cells (Fig. 5d−g). Together these data suggest a mechanism for MeCP2 regulation of miRNA production that holds true for both mouse *Rncr3* and human *RNCR3* in NEPCs/NPCs. We propose that (1) MeCP2 recognizes and binds to methylated cytosine sites on exons 2/3 of *Rncr3*; (2) MeCP2

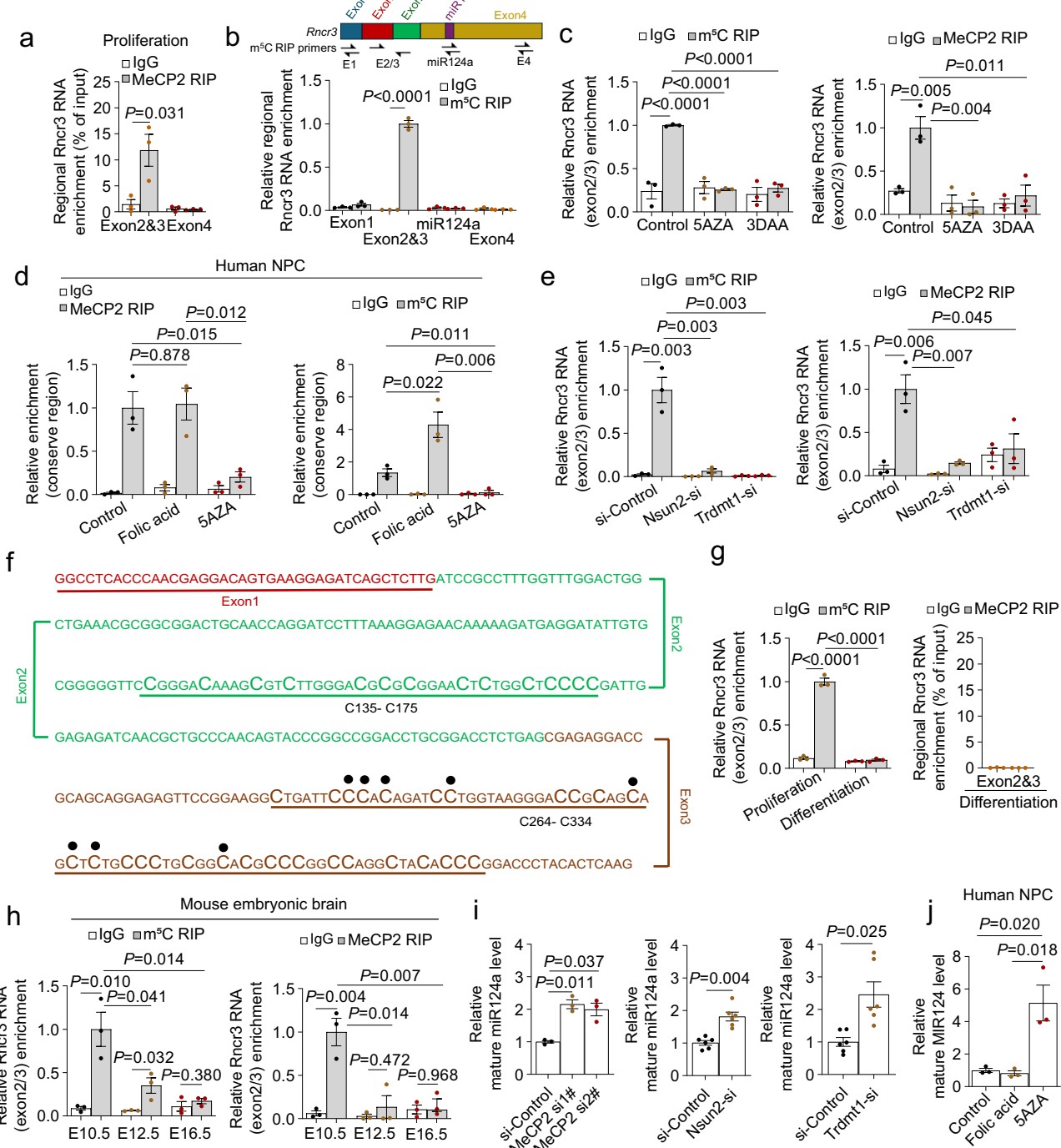

**Fig. 4 | MeCP2 protein binds cytosine methylated *Rncr3* exons 2/3. a** RNA immunoprecipitation (RIP) in NE-4C using anti-MeCP2 antibody or IgG control shows binding of *Rncr3* exons 2/3 by MeCP2 under proliferation conditions. **b** RIP with anti-5-methylcytosine antibody to capture methylated RNAs (m⁵C-RIP) probed for each region of *Rncr3* in proliferating NE-4C cells. **a, b** One-way ANOVA, mean ± s.e.m. (*n* = 3 biologically independent experiments). **c** Relative enrichment of *Rncr3* exons 2/3 region by m⁵C-RIP or MeCP2-RIP following treatment with methylation inhibitors 5AZA and 3DAA. **d** m⁵C-RIP or MeCP2-RIP with RT-qPCR data of human ReNcell CX cells show *RNCR3* RNA conserved region (Supplementary Fig. 1a, aligned to *Rncr3* exon 3) is methylated, methylation is increased by folic acid treatment, and 5AZA treatment disrupts methylation and MeCP2 binding. **e** Relative enrichment of *Rncr3* exons 2/3 as in (**c**), following knockdown of RNA methyltransferases *Nsun2* and *Trdmt1*. **c**–**e** Student's *t* test, two-sided; mean ± s.e.m. *n* = 3 biologically independent experiments. **f** Schematic representation of data for MeCP2-RIP Bisulfite Sequencing and total RNA Bisulfite Sequencing from NE-4C

cells. Capitalized C indicates methylated cytosine sites in exons 2 and 3 regions. Methylated nucleotides 271-313 were identified as the preferred region of MeCP2 binding (noted by black dots on the specific capitalized C). **g** Relative enrichment of *Rncr3* exons 2/3 upon NEPC differentiation (one-way ANOVA, mean ± s.e.m. *n* = 3 biologically independent experiments). **h** In vivo tests of relative enrichment of *Rncr3* exons 2/3 region by m⁵C-RIP or MeCP2-RIP from embryonic mouse brains at E10.5 (six brains pooled together for each experiment), E12.5 (six embryonic cortexes pooled together for each experiment) and E16.5 (three embryonic cortexes pooled together for each experiment); *n* = 3 biologically independent experiments. **i** Mature *miR124a* levels after knockdown of *Mecp2* (*n* = 3 independent experiments), *Nsun2* (*n* = 6) or *Trdmt1*(*n* = 6) in NE-4C cells under proliferation conditions. **j** Mature *MIR124-1* levels in human NPCs following folic acid and 5AZA treatment (*n* = 3). **h**–**j** Student's *t* test, two-sided; mean ± s.e.m. Source data are provided as a Source Data file.

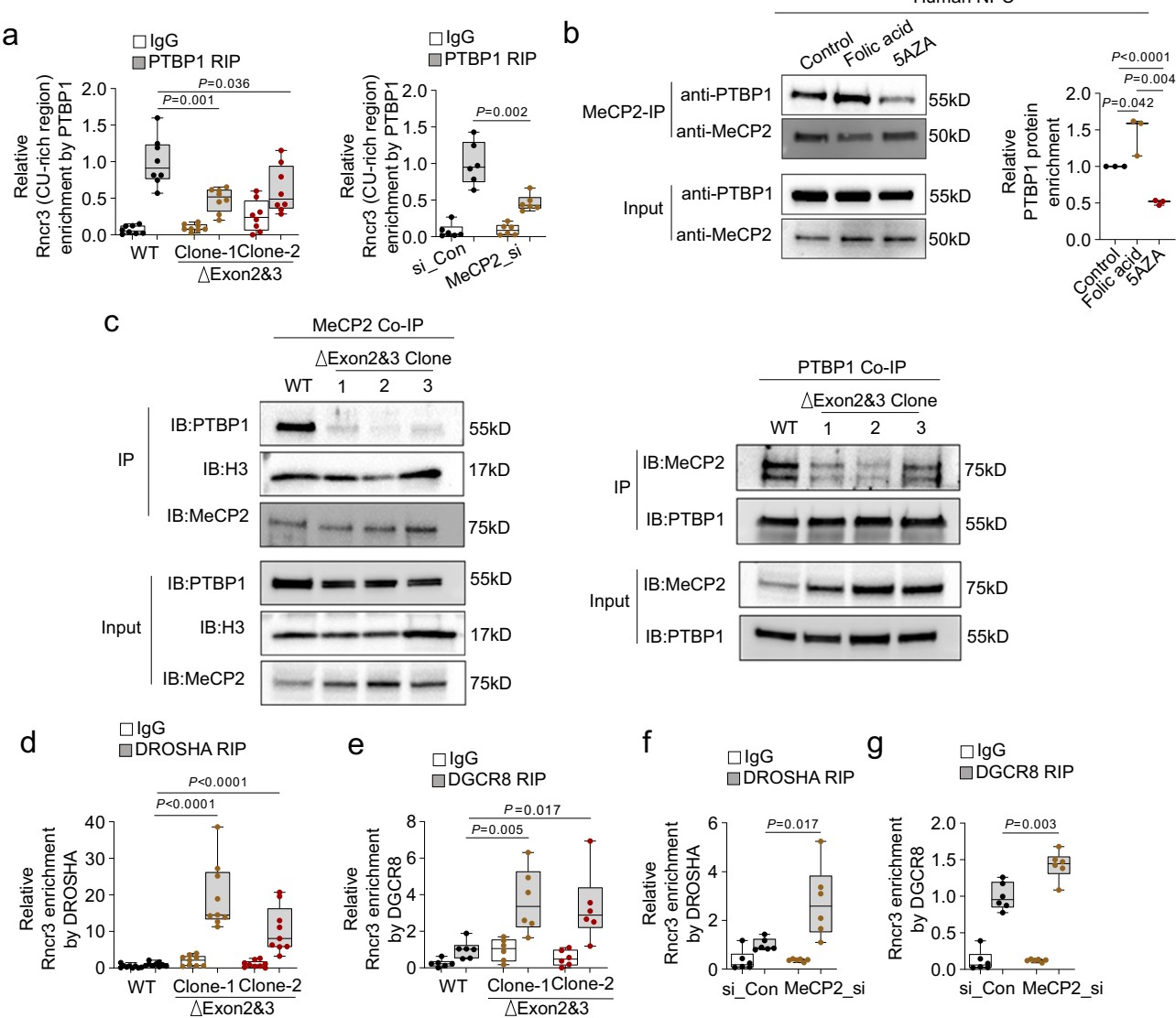

**Fig. 5 | MeCP2 recruits PTBP1 to restrict *miR124a* production. a** RIP data shows decreased binding between PTBP1 and *Rncr3* exon 4 CU-rich region upon exons 2/3 deletion ($n = 8$ biologically independent samples) or MeCP2 knockdown ($n = 6$) in NE-4C cells. **b** In human ReNcell CX cells, co-IP data for physical interaction between MeCP2 and PTBP1 in response to folic acid or 5AZA treatment ($n = 3$). Representative western blot on left, quantification on right. **c** Two-way co-IP data show physical interaction between MeCP2 and PTBP1, which is dependent on the presence of *Rncr3* exons 2/3. Two independent experiments were repeated with similar results. RIP data shows increased binding of *Rncr3* exon 4 *miR124a* region by DROSHA (**d**, $n = 9$ and **f**, $n = 6$) or DGCR8 (**e**, $n = 6$ and **g**, $n = 6$) in exons 2/3 deletion NE-4C cells or upon MeCP2 knockdown, respectively. **a**, **b**, **d**–**g** Student's *t* test, two-sided; boxplots show all data points with box which extends from the 25th to 75th percentiles and whiskers showing min to max, and the line in the middle of the box is plotted at the median. Source data are provided as a Source Data file.

recruits PTBP1 through protein-protein interactions; (3) PTBP1 recognizes and binds to a conserved CU-rich sequence adjacent to the 5′ of *miR124a* stem–loop within exon 4 of *Rncr3*; and (4) this complex prevents access of the miRNA processing machinery to restrict *miR124a* production.

**Conserved lysine residues in MeCP2-ID domain recognize methylated cytosines on RNA**

As a final set of mechanistic studies, we set out to determine how MeCP2 binds to methylated *Rncr3*. Two RNA binding domains (RBDs) have been defined for MeCP2 but these are poorly explored and have not been associated with recognition of cytosine methylated RNA: a lysine-rich domain within the NCoR1/2 interaction domain (human protein NP_001104262.1: NID; residues 285-309) and an arginine and glycine-rich (RG) domain in the intervening domain (ID, residues 174-219) which is C-terminal to the methyl-binding domain (MBD; residues 92-161; Fig. 6a)[44,57]. Deletion of the MBD domain, which is well-studied for its role in binding cytosine methylated DNA, is not required to bind *Rncr3*, whereas deletion of the RG region of the ID abolished *Rncr3* binding (Fig. 6b, Supplementary Fig. 7b, c). Within the RG region is a stretch of lysine residues which are highly conserved from human to fish (Fig. 6a: residues 183–192 in human, 171–180 in mouse) although potential functional significance is unknown. The protein ALYREF which binds m5C methylated mRNA through lysine 171 has been aligned with methyl DNA binding proteins[25] and this shows partial conservation to the MeCP2 ID lysine-rich region (Fig. 6a and Supplementary Fig. 7d). To test whether MeCP2 binding to m5C RNA requires these lysine residues, we created a full-length mouse MeCP2 expression construct with mutation of conserved lysines 171, 174 and 175 to alanines (K171A/K174A/K175A; Supplementary Fig. 7e). These lysines are needed for MeCP2 to bind cytosine-methylated RNA oligo-271 (Fig. 6c and Supplementary Fig. 7f). RIP from NE-4C cells transfected

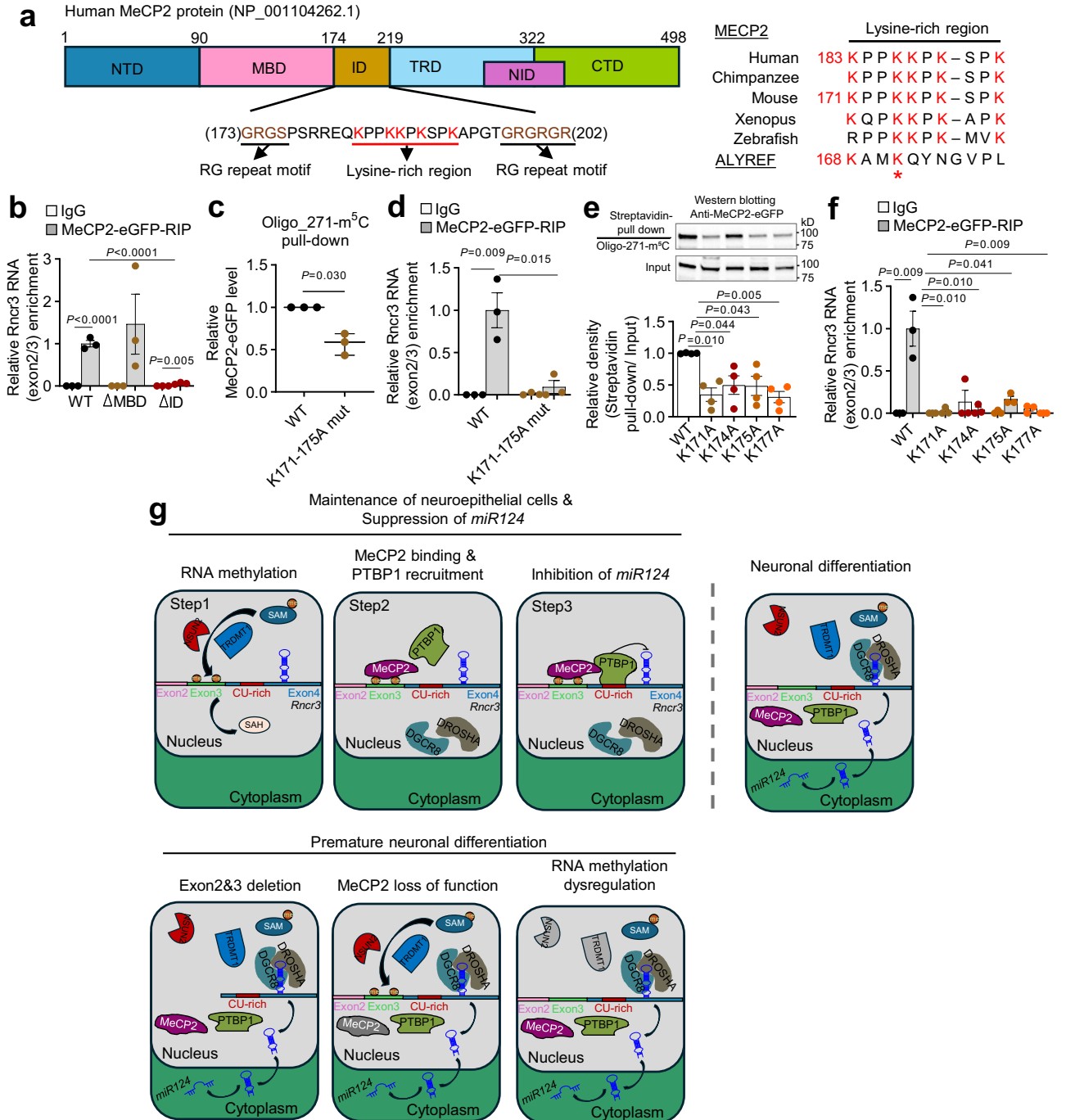

**Fig. 6 | MeCP2 recognizes cytosine methylated *Rncr3* RNA through conserved lysine residues in the intervening domain. a** Left: domain structure of MeCP2 emphasizing the intervening domain RG repeats (underlined) implicated in RNA binding[44] and conserved lysine rich region (underlined with lysines indicated in red) tested here for binding to cytosine methylated RNA. Right: protein sequence alignment of MeCP2 (mouse, NP_034918.1; human, NP_001104262.1) across species highlights five conserved lysine residues in the ID domain. Alignment with protein ALYREF highlights K171 (red star) which is essential for ALYREF binding to mRNA m⁵C sites[25]. **b** RIP data shows association of *Rncr3* exons 2/3 with MeCP2-eGFP-WT and MeCP2-MBD deletion but not the ID deletion (NP_034918.1, 161-190, see also

Supplementary Fig. 7b and c). Quantification of western blot assays for binding of methylated RNA Oligo-271-m⁵C by MeCP2-eGFP-wild-type (WT), MeCP2-eGFP-triple lysine mutation (**c**) and individual lysine mutations (**e**). **d, f** RIP data for association of *Rncr3* exons 2/3 with MeCP2-eGFP-WT and lysine-mutant proteins in NE-4C cells. **g** Model for methylated *Rncr3*-MeCP2 interaction in restricting *miR124a* processing in NEPC/NPC. In (**b**) (*n* = 3 biologically independent samples), (**d**) (*n* = 3), (**e**) (*n* = 4) and (**f**) (*n* = 3), student's *t*-test, two-sided; mean ± s.e.m., in (**c**) (*n* = 3), student's *t* test, two-sided; boxplots show all data points with box which extends from the 25th to 75th percentiles and whiskers showing min to max, and the line in the middle of the box is plotted at the median. Source data are provided as a Source Data file.

with eGFP-tagged MeCP2 constructs shows that *Rncr3* exons 2/3 associate strongly with wildtype MeCP2, but interaction dramatically decreases for the K171A/K174A/K175A MeCP2 mutant (Fig. 6d and Supplementary Fig. 7g). Single lysine mutations of MeCP2 also abolished interaction with methylated oligo 271 or *Rncr3* exons 2/3

(Fig. 6e, f, Supplementary Fig. 7h, i). These data provide insight for the role of MeCP2 as a methylated RNA reader protein and expand the relevant domains of MeCP2, highlighting lysines in the 171–180 region (human 183–192) as a RNA binding domain that specifically recognizes cytosine methylated RNA.

## Discussion

Our studies highlight a new paradigm of opposing activities of two non-coding RNAs derived from the same locus, *Rncr3* and *miR124a*. LncRNA *Rncr3* acts as a dual function switch centered on exons 2/3 for RNA cytosine methylation and NEPC/NPC proliferation and the suppression of *miR124a* expression until the onset of neuronal differentiation. Differential temporal expression of cytosine methylated *Rncr3* exons 2/3 versus *miR124a* coordinates the balance between expansion of the neuroepithelial progenitor pool with the promotion of neuronal differentiation, which is critical for ensuring the appropriate populations of neurons and glia in the central nervous system, especially the neocortex in mammals[28–31,33,35]. Deletion of *Rncr3* exons 2/3 in a NEPC model and in vivo mouse embryonic brain, in conjunction with disruption of human RNCR3 in NPCs and RNA expression data indicate the critical independent function of *Rncr3* in NEPC/NPC biology, beyond being the host transcript for *miR124a*. Our data highlight how one conserved non-coding locus can orchestrate brain development in an intragenic manner spanning early to late embryonic stages.

Deletion of the entire *Rncr3* locus in mouse manifests as postnatal microcephaly and neurological phenotypes[35], whereas our data show that null mutation of *Rncr3* exons 2/3 is lethal before E8.5, and haploinsufficiency causes severe phenotypes during embryogenesis including microcephaly, loss of NEPCs/NPCs and premature neuronal differentiation. Mechanistically, either a reduction of the neural progenitor pool or premature neuronal differentiation can lead to microcephaly[58,59]. Specific deletion of *Rncr3* exons 2/3 reduces NEPC/NPC proliferation and survival, while at the same time results in disinhibition of *miR124a* which promotes premature neuronal differentiation, further exacerbating depletion of the NEPC/NPC pool. In contrast, deletion of the entire *Rncr3-miR124a* locus removes both opposing activities – NEPC/NPC maintenance and neuronal differentiation—which we propose results in the milder phenotype in the full-length KO[35] of this dual regulator. Reintroduction of *miR124a* did not rescue the corticogenesis deficit and small brain size in the *Rncr3-miR124a* deletion mouse model, which was suggested to be due to the inability to recapitulate the timing or levels of *miR124a*[35]. Instead, our data are consistent with a different interpretation highlighting the independent function of *Rncr3* in NEPC/NPC biology and restriction of *miR124a* expression, especially in the neocortex.

Our RNA-seq comparison of exons 2/3 deletion versus full-length deletion cells provides insight into the molecular function of the *Rncr3-miR124a* dual function RNA. In both deletion lines there is dysregulation of genes that control cell proliferation and survival, although it remains to be determined whether these are direct or indirect *Rncr3* targets. In exons 2/3 deletion cells, in which *miR124a* is upregulated, there is a significant downregulation of *miR124* predicted target genes including *Minar1*, *Shh* and *Cdk6*, which ties the *Rncr3-miR124a* RNA to the regulation of Notch and Hedgehog signaling and the G1 stage of cell division, to maintain neuroepithelial cell self-renewal and restrict neuronal differentiation[5,51–53,60,61]. *LINC00599* (*RNCR3*) knock-down in human brain-derived neural progenitor cells also significantly decreased proliferation and increased cell death and reduction in cytosine methylation increased *MIR124-1* level. The human *RNCR3* gene is located on chromosome 8p23.1, within the chromosomal 8p region associated with neurological disorders, including microcephaly[62–65]. This leads us to speculate that mutation of *RNCR3* could contribute to neurodevelopmental disorders in humans. Our studies raise the idea that manipulation of the temporal control of *RNCR3/MIR124-1* production could be a target for the prevention of disorders of neural progenitors.

We have defined a regulatory circuit between *Rncr3* exons 2/3, MeCP2 and PTBP1 in the control of processing of *miR124a* in mouse and human NEPCs/NPCs. Deletion of mouse *Rncr3* exons 2/3 leads to a dramatic increase in mature *miR124a* and premature neuronal differentiation. Conserved cytosines in *Rncr3* exons 2/3 are methylated in proliferating NEPCs/NPCs (model Fig. 6g, step 1) and this RNA methylation permits recognition and binding by MeCP2 (step 2). A mechanism is highlighted whereby MeCP2 suppresses *miR124a* production by direct binding to methylated exons 2/3 of *Rncr3* and recruitment of PTBP1 to a conserved CU-rich region upstream of *miR124a* sequences (step 3). The regulatory complex of *lncRncr3*/MeCP2/PTBP1 blocks the miRNA processing machinery to limit the expression of exon 4 embedded *miR124a* and maintain NEPCs/NPCs proliferation (step 3). This contrasts with an indirect role for MeCP2 in suppressing miRNA processing by interfering with the binding of DGCR8 to RNA targets and hence limiting the assembly of DROSHA/DGCR8 nuclear microRNA-processing machinery complex at the miRNA stem-loop[46,47]. Disruption of either exons 2/3, *Rncr3* cytosine methylation, MeCP2, or RNA methyltransferases, all lead to abnormally increased *miR124a* expression (Fig. 6g, bottom panels). In vivo, as neuronal differentiation proceeds, cytosine methylation of exons 2/3 is lost and this corresponds with loss of MeCP2 binding and increased *miR124a* expression (Fig. 6g, top right panel). Further studies are needed to determine whether the temporal decrease in *Rncr3* exons 2/3 methylation upon differentiation is an active or passive process. Our data demonstrate a functional interaction between MeCP2 and PTBP1, and their crucial regulation of processing of the most abundant miRNA during early brain development. It is noteworthy that *miR124a* sequences are highly conserved across fish to human, but exons 2/3 are specifically conserved in mammals. We speculate that the exons 2/3 mediated regulatory circuit that restricts *miR124a* in early NEPCs/NPCs serves as an evolutionarily step in neural progenitor pool expansion and higher brain function in mammals.

Finally, our studies provide insight for MeCP2 as a m⁵C epitranscriptomic RNA reader protein. The well-characterized 5mC DNA methyl binding domain is not required to bind *Rncr3*. Instead, our studies highlight lysines 171–180 in the intervening domain (human MeCP2 residues 183-192) as key for association with cytosine methylated *Rncr3* RNA. In support, examination of RettBASE shows nonsense and frameshift mutations of lysines and prolines in this small region as well as numerous mutations within the 174-219 ID region associated with Rett syndrome. We speculate that the RG-repeat RNA binding modules in the ID provides affinity for RNA (currently thought to have higher affinity for G-quadruplex or GC-rich dsRNA)[44,66] and that the lysine-rich region specifies binding to cytosine methylated RNA. Future mutational studies should help to distinguish whether the RG-repeat and lysine-rich regions of MeCP2 function cooperatively or independently in the recognition m⁵C RNA versus other RNAs. Moreover, bioinformatic re-analysis suggests that MeCP2 binds a broad range of cytosine methylated RNAs in the brain. The identification of key conserved lysines critical for m⁵C RNA binding expands the functional domains of MeCP2, thus opening new possibilities for clinical assessment and identification of MeCP2 target RNA substrates during nervous system development.

## Methods

### Animal

All experiments described in this article comply with the relevant ethical regulations and all animal procedures were performed according to animal welfare guidelines and regulations approved by an IACUC approved protocol (Protocol #2590) at the University of Colorado Boulder.

Animals were kept under the standard conditions in filter top cages at 22 ± 2 °C, 55 ± 10% humidity with a 12-h light–dark cycle (6:00–18:00). Timed matings were performed to obtain somite and age matched embryos. Both male and female mouse embryos were used in this study. C57BL/6 J mice were purchased from Jackson Labs and maintained as a breeding stock at the University of Colorado Boulder in accordance with an IACUC approved protocol.

**NE-4C cell line.** NE-4C cells (Supplementary Table 1) were derived from E9 mouse embryonic brain. Cells were grown at 37 °C and 5% CO2 under proliferation conditions in MEM (Invitrogen) supplemented with 5% FBS, 1×MEM nonessential amino acids (Invitrogen) and 1×GlutaMax (Invitrogen). Neuronal differentiation was induced by the addition of 1 μM all-trans retinoic acid (ATRA, Supplementary Table 2) for 24−48 h. For folic acid experiments, the cells were treated with folic acid (Supplementary Table 2) at concentrations of 0.026 mg/L up to 100 mg/L in NE-4C growth medium (1 mg/L or 2.6 mg/L is physiological concentration)[67]. The NE-4C cell line is routinely tested in the lab for mycoplasma contamination.

**ReNcell CX immortalized cell line.** The ReNcell CX immortalized cell line (Supplementary Table 1) was originally derived from the cortical region of human fetal brain tissue and it retains the ability to differentiate into neurons and glial cells. The cells were maintained as neural stem cells on laminin coated tissue culture plates in NSC media (Millipore, SCM005) containing epidermal growth factor (20 ng/mL) and basic fibroblast growth factor (20 ng/mL, Supplementary Table 3). Neuronal differentiation was induced by withdrawing epidermal growth factor and basic fibroblast growth factor. In the regular culture media for ReNcell CX cells, folic acid concentration is 6 μM, therefore, for the folic acid experiments, the cells were treated with 6 μM (control) or 12 μM total folic acid.

**Genome editing using CRISPR/Cas9.** The CRISPR-Cas9 DNA editing experiment was adapted from[68,69]. Briefly, a pair of complementary oligos for each CRISPR targeting sgRNA was annealed with 5′ overhangs of 'ACCG' and 'AAAC'. The annealed DNA was inserted into a BsaI-linearized pGL3 vector with the U6 promoter. SgRNA sequences (Supplementary Table 4) were designed to target specific genomic sites for *Rncr3* full length deletion or around the region of exon 2 to exon 3. NE-4C cells were transfected with 0.5 μg Cas9 and two sgRNA plasmids (1 μg) by Lipofectamine 3000 (Supplementary Tables 2, 5) one day after plating $0.5 \times 10^5$ cells per well in a 24-well plate. On Day 2 post-transfection, puromycin (2 μg/ ml, Supplementary Table 2) was added and cells cultured for an additional 2−4 days. After puromycin selection, single cells were seeded into individual wells of 96 well plates by limiting dilution. Genomic DNA was isolated for genotyping PCR screening (Supplementary Table 6) followed by Sanger sequencing to confirm different deletion types (*Rncr3* exons 2/3 deletion or *Rncr3* full-length deletion).

In the exon 2&3 deletion line, the resulting RNA transcript includes a cryptic exonic fragment between exons 1 & 4 (Supplementary Fig. 3b−e). To clarify any negative or positive effects of this cryptic exon on *miR124a* biogenesis, we created different constructs to express wildtype *Rncr3*, *Rncr3* exon2/3 deletion with the cryptic exon (Δexon2&3 + ) or without the cryptic exon region (Δexon2&3; Supplementary Fig. 3c-e). The different plasmids were transfected into *Rncr3* KO NE-4C cells to assay the level of *miR124a* production. The results (Supplementary Fig. 3f) show full-length *Rncr3* (containing exons 2&3) has limited *miR124a* expression (17.5 fold over background in the KO); while the two plasmids of *Rncr3* Δexon2&3 and Δexon2&3+ induced *miR124a* by 92-fold and 88-fold, respectively. This indicates the cryptic exon does not alter functionality.

**Rncr3 exons 2/3 deletion mice.** The same sgRNA design strategy as for CRISPR/Cas9 genome editing in cells was used to target exons 2/3 of mouse *Rncr3*. The Mouse Genetics Core Facility at National Jewish Health (Denver, CO) injected the guide RNA and Cas9 protein into FVB/N zygotes (Supplementary Table 7). Twenty-one pups were obtained and genotyping showed five individuals that were positive for the exons 2/3 regional deletion, 2 males and 3 females. The sequence data showed the mutants harbored the expected 1398 bp deletion spanning exons 2/3 of *Rncr3*. The genotyping test utilized a nested-PCR system,

and the primers are shown in the Supplementary Table 6. The original line has been maintained on an FVB/N background. In addition, FVB/N (FVB) *Rncr3*^Δexons2/3/WT^ heterozygous mice were outcrossed to C57BL/6 (C57) mice and the resulting FVB/C57 offspring were outcrossed for at least 5 more generations to C57 to yield >F6 progeny of essentially C57 background. These animals on the C57 background were used for all experiments performed in this manuscript, both male and female mouse embryos were used in this study.

**Whole-embryo culture.** As described in our previous publications[67,70,71], somite-staged E8.5 embryos were dissected and placed in culture media with scrambled antisense oligonucleotide (ASO) or ASO directed against *Rncr3* (final concentration 100 nM, Supplementary Table 8) within a temperature- and gas-controlled incubator (37 °C, 5% O2, 5% CO2) for culturing ~12 h. Culture medium was DMEM without Phenol Red and contained 55 U/ml penicillin/ streptomycin, 2.2 mM glutamax, 11 mM Hepes buffer, mixed with 75% rat serum (Supplementary Table 3). In Supplementary Fig. 2i, E9.0 embryos after 12 h culture were fixed for 20 min at room temperature in 4% PFA and then washed in 1X PBS three times for 5 min and processed for immunofluorescent staining.

**Analysis of coding potential.** The Coding-Potential Assessment Tool (http://lilab.research.bcm.edu/cpat/index.php)[72] was used to predict the protein-coding potential of *Rncr3*. CPAT uses a logistic regression model built with four sequence features: open reading frame size, open reading frame coverage, Ficket TESTCODE statistic and hexamer usage bias. The Ficket score is independent of the ORF, and when the test region is ≥200 nt in length (which includes most lncRNA), this feature alone can achieve 94% sensitivity and 97% specificity, with 'no opinion' on 18% of the sequences. Hexamer score determines the relative degree of hexamer usage bias in a particular sequence. Positive values indicate a coding sequence, whereas negative values indicate a noncoding sequence.

As a second method, we searched the ribosome footprinting RNA-seq database of published literature[50].

**Quantitative real-time PCR assays.** For quantitative real-time PCR assays, total RNA was extracted from embryos or cells using *Quick*-RNA™ Miniprep Kit (Supplementary Table 9). For the measurement of *Rncr3* and coding genes transcripts, total RNA was used for reverse transcription. cDNA was synthesized using Oligo d(T)₂₃VN primer (NEB, #S1327) and 1 μg of purified RNA by First Strand cDNA Synthesis Kit (Supplementary Table 9). Quantitative real-time PCR was performed with the LightCycler® 480 Real-Time PCR System (Roche). Results were normalized to 18 S or *Hprt* (in mouse embryos and NE-4C cells) or GAPDH in human ReNcell CX cells. Data analysis used the comparative CT method in Office Excel software. Mature microRNAs were specifically detected using a modified version of the stem-loop RT-PCR protocol described by[73]. For the measurement of mature miRNA levels, small fraction RNAs (<200 nt) were isolated with *Quick*-RNA™ Miniprep Kit and quantified with quantitative real-time PCR assays using the reverse transcription stem-loop primers. As a control, U6 small nuclear RNA was also amplified using stem-loop RT-PCR (Supplementary Table 10).

**miRNA expression manipulation.** NE-4C cells were transfected with either miRCURY LNA miRNA Mimics (Qiagen) or locked nucleic acid (LNA)-enhanced antisense miRNA inhibitors targeting miR-124a-3p. Cells were harvested 48 h later, and small fraction RNAs (<200 nt) were isolated with *Quick*-RNA™ Miniprep Kit and quantified with quantitative real-time PCR assays using the reverse transcription stem-loop primers. Cell proliferation and cell survival were assessed 2 days after transfection in proliferation conditions and differentiation denoted by TUJ1 staining assessed 2 days after transfection in differentiation conditions.

**TargetScanMouse prediction of miR124a targets.** Using the *TargetScanMouse* website, we created a list of all potential target genes of *miR124a* (a total of 1580 genes). *Rncr3* exons 2/3 deletion leads to significant up-regulation of mature *miR124a* levels, hence we speculated that the levels of mRNAs that are regulated by *miR124a* would show a downward trend. Therefore, the *miR124a* target list was compared with the list of 878 down-regulated genes in exons 2/3 deletion cells. The number of overlapping genes between the two lists was 117. We excluded genes that were downregulated in both exons 2/3 deletion cells and *Rncr3* full length KO cells (16 genes). The remaining 101 *miR124a* target genes that were only downregulated in exons 2/3 deletion cells and remained unchanged in *Rncr3* full length KO cells are shown in Fig. 2e, and the specific gene information is shown in the sheet Fig 2e in Source data_Fig. 2.

**Immunofluorescent staining and TUNEL assays.** Embryos, embryonic coronal sections or cultured cells were fixed for 10-15 min at room temperature in 4% PFA, blocked for 1 h at room temperature, and stained with primary antibody (Supplementary Table 11) overnight at 4 °C. DAPI or Hoechst (1:1000 dilution) was used to label and count number of nuclei.

In Situ Cell Death Detection Kit (Supplementary Table 9) was used for TUNEL assays. Embryos or cultured cells were fixed by 4% paraformaldehyde for 10-15 min at room temperature, then a TUNEL labeled assay was performed following the manufacturer's instructions.

Imaging was performed on a Nikon A1 confocal microscope. IMARIS 8.0.2 and ImagJ 1.53c were used to perform quantification analysis. Double-blinded manner was performed for data collection.

**RNA in situ hybridization.** Whole-mount in situ hybridization of *Rncr3* and *miR-124a-3p* was performed with digoxigenin-labeled antisense RNA probes (Supplementary Table 12) for *Rncr3* as described in previous publications[28,35] and following manufacturer's specifications of miRCURY LNA miRNA ISH Optimization kit handbook (Qiagen). Digoxigenin-labeled riboprobes were synthesized by SP6 RNA polymerase using the linearized plasmid as a template in the presence of 11-digoxigenin UTPs (Supplementary Table 2).

For *miR124a* detection, we used a *hsa-miR-124-3p* miRCURY LNA miRNA Detection Probe (PO HPLC, 5'-DIG; Qiagen, YD00619867-BCE). Hybridized probes were visualized using anti-DIG-alkaline phosphatase (AP) conjugate and BM Purple AP substrate (Roche) as recommended.

**ASOs and siRNA transfection in cells.** NE-4C cells were seeded in 24-well plates at $1 \times 10^5$ per well in 0.5 ml medium and after 24 h transfected with 20 nM of 2′-O-Methyl (2′OMe) RNA "gapmer" antisense oligonucleotides (IDT, Supplementary Table 8) using Lipofectamine 2000. Human ReNcell CX cultures were transfected with a ASO mixture (four 20-22 nt oligonucleotides, Supplementary Table 13) at 1 µM and 2 µM. Mouse *Mecp2*, *Nsun2*, *Trdmt1* and *Mettl3* mRNAs were knocked down using corresponding Silencer® siRNAs (Thermofisher; siRNA ID: 156281, 68154; 186409, 186410; 161529, 161530; 74203, 74109). In RNAi experiments, the Silencer® Negative Control #1 siRNA (AM4611) was used as a negative control.

**5-AZA and 3DAA treatment.** We used the demethylating agents 5-azacytidine (5-AZA, Supplementary Table 2) and 3-deazaadenosine (3DAA). 5-AZA was dissolved in phosphate-buffered saline as 10 mM stock, filtered (0.22 µM), and stored at −20 °C in aliquots that were thawed immediately prior to use at 10 µM, 5 µM, 3 µM and 1 µM to treat NE-4C cells. 3DAA was dissolved in DMSO as a 50 mM stock, and the working concentration was 200 µM on NE-4C cells. Given the short half-life of the drugs in culture media, 5-AZA or 3DAA medium was prepared and replaced daily. For human ReNcell CX cells we used a final concentration of 5 µM 5-AZA.

**RNA Immunoprecipitation (RIP) Assay.** RIP was adapted from the manufacturer's specifications of RNA ChIP-IT Magnetic Chromatin Immunoprecipitation Kit (Supplementary Table 9). Briefly, $1–2 \times 10^7$ cells were used per RIP experiment. Cells were treated with 1% formaldehyde to cross-link in vivo protein-RNA complex for 10 min, and glycine was added to stop the cross-linking. Cells were washed and lysed by lysis buffer (with 40 U/ml RNase inhibitor, and 5 µl protease inhibitor cocktail and 5 µl PMSF) on ice for 30 min, then were centrifuged for 10 min at 3000 × *g* in a 4 °C microcentrifuge. The nuclei pellet was resuspended in ice cooled shearing buffer and the chromatin was sheared by using a sonicator with an optimized chromatin shearing condition (optimal sonication shearing should result in a 100–1000 bp smear). Nuclear membrane and debris were pelleted by centrifugation at 20,000 × *g* for 10 min in a 4 °C microcentrifuge. The sonicated lysed material was treated with DNase I before performing the RIP experiment. To 10 µl Input sample was added 88 µl RNA IP buffer, 0.5 µl RNase Inhibitors and 2 µl 5 M NaCl, (stored at −80 °C). RIP reactions were performed by adding pre-washed protein G magnetic beads (25 µl per reaction), RIP buffer, 15 µl, sheared material (-100 µl), RNase inhibitor (100 U/ml), protease Inhibitor cocktail, with the reaction components kept on ice during mix preparation. Anti-MeCP2, PTBP1, DROSHA, DGCR8 and eGFP antibodies (4 µg per RIP reaction) were then added to a total reaction volume of 150 µl and incubated on a rotator overnight at 4 °C. RNA complex bound beads were washed four times with ice cold 200 µl wash buffer 1 and two times with ice cold 200 µl wash buffer 2, removing as much supernatant as possible without disturbing the beads. 100 µl elution buffer was added to the beads and the pellet resuspended by rotation for 30 min at room temperature. A magnet was used to pellet the beads and the supernatant transferred into a fresh tube to which 2 µl 5 M NaCl and 2 µl proteinase K was added. The Input control sample was treated with 2 µl proteinase K. The samples were incubated at 42 °C for 1 h to digest the protein and then incubated for 1.5 h at 65 °C to reverse the cross-link. RNA was purified with TRIzol (Supplementary Table 2) and followed by DNase I treatment. RNA was then quantified, and equal amounts were analyzed via qRT-PCR.

**m⁵C RIP.** Methylation-RNA immunoprecipitation (meRIP) was adapted from publications[56,74]. Isolated and purified RNA was randomly fragmented by incubation at 70 °C for 15 min using Fragmentation Reagent (Supplementary Table 2) and the fragmentation stopped by adding Stop Solution. The fragmented RNA (10 µg) was incubated with anti-m⁵C antibody (2 µg per IP, Supplementary Table 11) in 100 µl IP buffer (25 mM Tris-HCl pH 7.4, 150 mM KCl, 5 mM EDTA, 0.5 mM DTT, 0.5% Igepal CA-630 (v/v), RNase Inhibitors, and vanadyl ribonucleoside complex) on an end-to-end rotator for overnight at 4 °C. The same procedure was performed for a control reaction using Rabbit IgG (Supplementary Table 11). The protein A/G magnetic beads slurry (Supplementary Table 3) was washed twice with IP buffer and the beads were resuspended in IP buffer supplemented with BSA (0.5 mg/ml) on a rotating wheel for 1 h at 4 °C, followed by magnet pelleting of the beads and washing of the beads with IP buffer three times. 30 µl pre-washed protein A/G magnetic beads were added to the fragmented RNA and antibody treated samples, incubated for 2 h on a rotator at 4 °C, followed by washing the beads-antibody complexes four times with washing buffer I (50 mM Tris-HCl, pH 7.4, 1 M NaCl, 1% NP-40, and 1% sodium deoxycholate), and two times with washing buffer II (50 mM Tris-HCl, pH 7.4, 1 M NaCl, 1% NP-40, 1% sodium deoxycholate, and 1 M Urea). Co-precipitated RNAs were isolated by resuspending beads in 1 ml TRIzol and purified by phenol/isopropanol precipitation. RNA was then quantified, and equal amounts were analyzed by qRT-PCR.

**Co-immunoprecipitation.** As described in our previous publication[67] and the manufacturer's specifications of Nuclear Complex Co-IP Kit

(Supplementary Table 9), cells (1–2 × 10$^7$) were collected and nuclear extracts prepared. Anti-MeCP2 or anti-PTBP1 antibodies (5 μg per co-IP reaction) were added into 500 μg nuclear extract and rotated overnight at 4 °C. Washed PureProteome Protein A/G Mix Magnetic Beads (50 μl) were added and samples incubated at 4 °C for 1 h on a rotator. Beads were washed five times in ice-cold wash buffer before elution in 2× Laemmli buffer at 95 °C for 10 min. Protein interactions were subsequently assayed via western blot and the data quantified using ImageJ.

**Re-analysis of published MeCP2-RIP-Seq and nuclear poly(A) RAN-bisulfite-Seq datasets.** To provide an indication of the overlap between MeCP2-bound RNAs and m$^5$C methylated RNAs, we re-analyzed two published databases[43,56]. Both datasets are derived from the brain tissue of adult mice, and hence the two datasets should be relatively comparable. By comparing the results of MeCP2-RIP-Seq and the results of nuclear poly(A) RAN-bisulfite-Seq, we obtained a subset of target mRNA molecules bound by MeCP2 protein that can be modified by m$^5$C methylation in the mouse brain (a total of 1054 mRNAs). Since the original poly(A) RAN-bisulfite Seq study of the Thomas, A. et al database[56] provided a total of 1511 m$^5$C mRNAs, we calculated that the subset of target mRNA molecules bound by MeCP2 protein and that can be modified by m$^5$C methylation accounts for about 70% of the total m$^5$C mRNA. The Source data for Supplementary Fig. 4k lists the exact position of each methylated cytosine in the mRNA subset.

**Dot blots.** Dot blot analysis of m$^5$C levels was adapted from a previous publication[75]. Briefly, using proliferating NE-4C cells, purified RNA from MeCP2-eGFP pulldown samples (1 μg) or input samples (5 μg) were denatured by heating at 95 °C for 3 min, followed by chilling on ice immediately. RNA was spotted on a Hybond-N+ membrane (Amersham) and crosslinked at 80 °C for 30 min. The membrane was washed with PBST buffer (RNase free PBS with Tween-20), then blocked by 5% non-fat milk in PBST for 1 h and incubated with anti-m$^5$C antibody (1:500) overnight at 4 °C. The membrane was washed for 5 min three times in PBST buffer, and then incubated with a secondary antibody, HRP-conjugated Goat anti-mouse IgG (1:2000) for 1 h at room temperature. After washing by PBST buffer for 5 min three times, the enzyme substrate was added to the membrane and incubated for 5–10 min. The secondary antibody signal was visualized using a chemiluminescence kit according to the manufacturer's instructions.

**ChIRP-western blot.** Two 15 cm dishes of NE-4C cells were used per ChIRP-Western blot assays. Cell harvesting, lysis, disruption, and ChIRP were essentially performed as the standard procedures described in[76]. NE-4C cells were cross-linked in 3% formaldehyde for 30 min, followed by 0.125 M glycine quenching for 5 min. Lysis buffer (50 mM Tris-Cl pH 7.0; 10 mM EDTA; 1% SDS; with fresh Protease Inhibitor (PI), PMSF and RNase inhibitor) was added to the cell pellet (1 mL lysis buffer for 100 mg pellet) and the pellet resuspended. The cell lysate was sonicated in a Bioruptor in a 4 °C water bath at highest setting with 30 seconds ON, 45 seconds OFF pulse intervals, until no longer turbid. The sonicated samples were centrifuged at 20,000 × $g$ for 10 min at 4 °C. To 1 mL supernatant was added 2 mL hybridization buffer (750 mM NaCl; 1% SDS; 50 mM Tris-Cl pH 7.0; 1 mM EDTA; 15% formamide), and then appropriate volume of probes added (100 pmol probe per 1 mL supernatant, Supplementary Table 14). The samples were mixed and incubated at 37 °C overnight with shaking. C-1 magnetic beads (100 μL; ThermoFisher Scientific, 65001#) were washed in 1 mL lysis buffer three times and the beads resuspended in original volume of lysis buffer (supplemented with fresh PMSF, P.I and RNase inhibitor). Washed beads (100 μL) were added to the hybridization reaction, mixed well and incubated at 37 °C for 30 min with shaking.

After ChIRP reactions, the beads were washed 5 times with wash buffer (2x NaCl and Sodium citrate; 0.5% SDS; with fresh PMSF). For protein elution, beads were collected by the magnet, resuspended in 40 μL 1 × laemmli sample buffer and boiled at 95 °C for 30 min, and the supernatant collected. Final protein samples were size-separated in 4–20% Mini-PROTEAN® TGX™ Precast Protein Gels (Bio-Rad) and used for western blot experiments.

**Bar-coded bisulfite sequencing.** Total RNA was isolated from NE-4C cells and then used for RNA bisulfite sequencing following the steps of RNA-bisulfite treatment, poly (A) polymerase tailing reactions, bar-coded cDNA synthesis, nested PCR and Sanger sequencing. (1) RNA-bisulfite treatment: Total RNA from NE-4C cells was incubated for two rounds with the EZ RNA Methylation Kit (Supplementary Table 9) according to the manufacturer's instructions (5 min at 70 °C followed by 60 min at 64 °C). (2) poly (A) polymerase tailing reactions (Supplementary Table 9): Bisulfite treated RNA was combined with the following reaction components on ice: 2 μL Poly(A) Polymerase 10X Reaction Buffer, 2 μL 10 mM ATP, 0.5 μL RNase Inhibitors, 2 μg of bisulfite treated RNA and 1 μL Poly(A) Polymerase (4 Units), and DEPC water to total reaction volume of 20 μL. Samples were incubated at 37 °C for 15 min, then EDTA added (final concentration of >11 mM) to stop the reaction. The poly (A) polymerase tailed RNA was purified with TRIzol and finally diluted with 10 μL DEPC water. (3) Bar-coded cDNA synthesis: Reverse transcription (adapted from SMARTer™ RACE cDNA Amplification Kit, Supplementary Table 9) of bisulfite treated and polymerase tailed RNA was performed with a specific bar-coded anchor RT primer: (AAGCAGTGGTATCAACGCAGAGTACNNNT(30)VN, and V = A, G or C; N = A, C, G or T; "NNN" noted as bar-code, Supplementary Table 15). This bar-coded cDNA synthesis strategy allows for tagging/bar-coding of $N^3$ different RNA molecules during cDNA synthesis. (4) Nested PCR (adapted from SMARTer™ RACE cDNA Amplification Kit, Supplementary Table 9): Two rounds of nested PCR reactions were performed with a series of primers specific for deaminated sequences along *Rncr3* exons 2/3 regions and the Universal Primer Mix (round 1 PCR) or Nested Universal Primer (Supplementary Table 15). The amplicons were cloned into pGEM-T Easy vector (Promega) followed by Sanger sequencing of 8-10 individual clones. Under neuronal differentiation conditions, we examined 50 clones from total RNA bisulfite sequencing to assess the methylation modifications in *Rncr3* exon3 region.

**RNA affinity chromatography-western blot assay.** The biotin-labeled RNA oligonucleotides (synthesized by IDT) with (Oligo- m$^5$C) or without m$^5$C (Oligo-C) sequence are as follows:

Oligo-271-m$^5$C/ Oligo-271-C: 5′-biotin-UGAUUC**X**CACAGAUCCUGGUAAG-3′, and negative control (NC) Oligo-323-m$^5$C/ Oligo-323-C: 5′-biotin-ACGCCCGGC**X**AGGCUACACCCGG-3′ (X = C or m$^5$C). HEK 293 cells were transfected with wild-type or mutated MeCP2-eGFP (1 × 10$^7$ cells per reaction, transfection efficiency >70%). In vivo RNA pull-down assays (adapted from[25]) were carried out using nuclear extracts that were precleared for 1 h at 4 °C by incubation with streptavidin-conjugated magnetic beads (NEB) in binding buffer (50 mM Tris-HCl pH 7.5, 250 mM NaCl, 0.4 mM EDTA, 0.1% NP-40, 1 mM DTT) supplemented with 0.4 U/μl Rnasin (Promega). Biotin-labeled RNA oligonucleotides (18 μg) were incubated with pre-cleared nuclear extracts (total volume = 120 μL) overnight at 4 °C under gentle rotation. Streptavidin-conjugated magnetic beads were pre-cleared by incubation with 0.2 mg/ml tRNA and 0.2 mg/ml BSA for 1 h at 4 °C under gentle rotation. Pre-cleared magnetic beads were incubated with biotin-labeled RNA oligonucleotides and pre-cleared nuclear extracts for 1 h at 4 °C under gentle rotation. Magnet pelleted beads were washed three times with wash buffer (50 mM Tris-HCl pH 7.5, 250 mM NaCl, 0.4 mM EDTA, 0.1% NP-40, 1 mM DTT, 0.4 U/μl Rnase Inhibitors). For protein elution, beads were magnet collected and

resuspended in 40 μL 1 × laemmli sample buffer and boiled at 95 °C for 10 min. The beads were pelleted by magnet and the supernatant subjected to protein size-separation in 4–20% Mini-PROTEAN® TGX™ Precast Protein Gels (Bio-Rad) and used for western blot experiments.

**Plasmids.** To express appropriate sgRNAs in NE-4C cells for deletion of *Rncr3* exons 2/3 or for *Rncr3* full length KO, the pGL3 vector (Supplementary Table 5) with the U6 promoter was used and co-transfected with pCAS9 into cells. pGL3-U6-sgRNA-PGK-puromycin was a gift from Xingxu Huang[77] (Addgene plasmid # 51133; http://n2t.net/addgene:51133; RRID: Addgene_51133) and pCas9_GFP was a gift from Kiran Musunuru[78] (Addgene plasmid # 44719; http://n2t.net/addgene:44719; RRID: Addgene_44719).

For in vivo RNA pull-down assays and MeCP2-eGFP RIP experiments, cells were transfected with plasmids expressing eGFP fused with wild-type (pEGFP-N1_MeCP2(WT), gift from Adrian Bird[57]; Addgene plasmid # 110186; http://n2t.net/addgene:110186; RRID: Addgene_110186) or other MeCP2 mutants (Supplementary Tables 16 and 17). Other MeCP2 mutations: (1) MBD deletion, in which 78 amino acids were deleted encompassing glycine 92 to C-terminal glutamic acid 169 (NP_034918.1). (2) ID deletion in which 30 amino acid residues (GRGSPSRREQKPPKKPKSPKAPGTGRGRGR) of the ID domain were deleted. (3) Lysine to alanine mutations in K171-K175 region (K171/174/175 A) in MeCP2 and individual mutations generated using Quik-Change II Site-directed mutagenesis kit (Agilent, #200521).

**MS2-tagged Rncr3 RNA affinity purification and western blot.** MS2-tagged *Rncr3* RNA affinity purification was adapted from publications[79–81]. Briefly, phage-CMV-*Rncr3*-24×MS2bs (Supplementary Table 5) was constructed by introducing an NotI-BsrGI fragment containing the wildtype full-length *Rncr3* gene into the corresponding sites in the backbone of phage-CMV-CFP-24×MS2bs, a gift from Robert Singer (Addgene plasmid # 40651; http://n2t.net/addgene:40651; RRID: Addgene_40651). The phage-CMV-*Rncr3*-24×MS2bs with single site mutation of 271 C > T or multiple C > T mutations (at positions 271/272/274/280/297/300/302/313) were generated by using QuikChange II Site-directed mutagenesis kit (Supplementary Table 9) with the corresponding PCR primers (Supplementary Table 18).

The CRISPR induced *Rncr3* full-length KO cells described above were co-transfected with wildtype or mutated phage-CMV-*Rncr3*-24×MS2bs plasmids and pMS2-GFP plasmid, a gift from Robert Singer (Addgene plasmid # 27121; http://n2t.net/addgene:27121; RRID:Addgene_27121), which expresses the MS2-GFP fusion protein. After 48 h, cells were treated with 1% formaldehyde to cross-link protein-RNA complex for 10 min, and glycine was added to stop the cross-linking. The next steps basically follow the above-described RNA Immunoprecipitation protocol. RIP reactions were performed by adding RIP buffer, sheared material, RNase inhibitor, protease Inhibitor cocktail and anti-MS2 antibody (4 μg per RIP reaction), then incubated on a rotator overnight at 4 °C. PureProteome Protein A /G Mix Magnetic Beads were added and samples incubated at 4 °C for 1 h on a rotator. Beads were washed total six times in ice-cold wash buffer 1 and wash buffer 2 before elution in 2× Laemmli buffer at 95 °C for 10 min. Interacting proteins were subsequently assayed via western blot and the data quantified by ImageJ.

## PolyA(+) RNA-seq and bioinformatics analysis

For sample preparation, three individual clones of CRISPR-Cas9 DNA edited *Rncr3* exons 2/3 deletion or *Rncr3* full-length deletion in NE-4C cells were used. Three biological replicates of non-manipulated NE-4C cells were used as wild type control. All cell samples were cultured under proliferation conditions for three days and then collected and subjected to total RNA extraction.

One μg of RNA was used for library preparation (Universal Plus mRNA-Seq with NuQuant, Tecan Genomics, #0520). Samples were run on the Illumina NovaSEQ6000 system at a depth of 80 million paired-end reads. Following de-multiplexing, fastQ files were assessed for quality using fastQC (version 0.11.8). Trimgalore (0.4.3) was used to remove adapters, jumping index content unique to Tecan library kit, and sections of reads with low quality with a minimum of 20 bp read length. Samples were then aligned using tophat2 (version 2.1.1)[82] utilizing bowtie2 (version 2.2.9)[83], beyond default options for a sensitive paired end alignment, reads were aligned in reverse to accommodate the first-strand-sense reads unique to the Tecan library prep kit. Samples were then sorted and indexed using samtools (version 1.10)[84]. Read counts were assessed using htseq (version 0.9.1 under python distribution 3.6.3)[85] and options were set for a paired-end reverse alignment. Differential expression was evaluated using deseq2 (version 3.14)[86] within Rstudio (version 1.4.1103) utilizing R (version 4.1.2). Deseq2 analysis was performed on all 9 samples simultaneously portioned by design based on transcript knockout length. Genes identified to be differentially expressed between wild-type and a mutant (multiple hypothesis testing *P*-value < 0.05 and log2fold change > 0.5 or < -0.5) were further submitted to a likelihood ratio test within Deseq to eliminate batch effects as a possible contributor to differential expression. GO enrichment analysis was performed using web server[87] and visualized by Rstudio (version 3.5.1) and excel[88], biological process enrichment with *P* value less than 0.05 was considered significant and visualized. UCSC Genome Browser on Mouse (GRCm38/mm10) and Human (GRCh38/hg38)[89] were referenced to sequences alignments in all data.

### Statistical analysis

One-way ANOVA and two-tailed Student's *t*-tests were used to analyze the experimental data. Values of visualized columns are represented as mean ± s.e.m.. For boxplots, midlines represent the median, boxes the interquartile range (25th to 75th percentile), and whiskers the range of data. *P* < 0.05 is considered as statistically significant. The statistical details can be found in the figure legends. Statistical analysis was performed using SPSS (25.0.0) and GraphPad Prism 9.0.0 (121). GraphPad Prism 9.0.0 (121) was utilized for data visualization.

### Reporting summary

Further information on research design is available in the Nature Portfolio Reporting Summary linked to this article.

## Data availability

The RNA-seq data generated in this study are available at GEO under accession GSE191133 (GEO Accession viewer (nih.gov)). Two published datasets were reanalyzed in the study (GSE38324: [https://doi.org/10.4161/rna.26921][43] and GSE83432: [https://doi.org/10.1186/s13059-016-1139-1][56]). Other data are available in the manuscript or the supplementary materials. Source data are provided with this paper.

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

## Acknowledgements

This work was supported by NIH R01NS110887 (L.A.N.) award. We thank Richard Davis and members of our lab, in particular David Engelhardt, Eric Jaffe and Jonathan Wilde, for experimental ideas and Drs. R. Davis, Justin Brumbaugh and Aaron Johnson for comments on the manuscript. We acknowledge the help and resources of the MCDB Light Microscopy Core.

## Author contributions

J.Z. and L.A.N. designed research. J.Z. and H.L. performed experiments and RNA sequencing and analysis. J.Z., H.L. and L.A.N. analyzed data and performed data interpretation. J.Z., H.L. and L.A.N. wrote the manuscript. H.L. prepared the figures. L.A.N. supervised the project.

## Competing interests

The authors declare no competing interests.
