## [Peer Review File · Nature Communications]

m5C methylated IncRncr3–MeCP2 interaction restricts miR124a-initiated neurogenesisREVIEWER COMMENTS

Reviewer #1 (Remarks to the Author):

The manuscript entitled "m5C methylated lncRncr3–MeCP2 interaction restricts miR124a-initiated neurogenesis" describes an interesting new regulation of noncoding RNA via the m5C modification and recruitment of MeCP2 therein. This study demonstrates how differential modification can alter RNA biogenesis of LINC00599 (RNCR3), lncRncr3 and miR124a-1 in neurogenesis. The authors show that inhibition of miR124a and genetic knockout of Rncr3 results in decreased NPC proliferation. Intriguingly, homing in on exons 2/3 of Rncr3 results in premature expression of miR124a and in turn, neuronal differentiation processes. The authors further pursue the mechanism of this regulation finding that MeCP2 binds to the exon 2/3 region of Rncr3 and not the miR124a region. Moreover, m5C modifications are specified in this same region. This specified site and modifications are differentially regulated during neuronal differentiation pointing to a functional relationship between m5C and MeCP2 being fine tuned to initiate neuronal differentiation/proliferation via biogenesis of miR124 and repression of miR124 prior to differentiation. Moreover, the authors find that MeCP2 binding to m5C recruits PTBP1 and inhibits miR124a production. Finally the authors identify where this recognition occurs on MePC2.

Together this demonstrates a fascinating new regulatory role of RNA modifications, to my knowledge the first time this has been described. The thorough mechanistic studies reveal even more surprises that DNA methyl binding region of MeCP2 is not required and recognition occurs elsewhere. Thus, there is a fascinating new post-transcriptional regulatory circuit that provides a clock for when miR124 is produced and timing of neuronal differentiation.

I could not find any flaws with the experimentation or interpretations of the data presented. Thus, I do not have any major suggestions or concerns to communicate to the authors.

The only thing I can think to ask is, how general this fascinating mechanism is? The authors could perform RIP or CLIP or fRIP Seq and determine how many other transcripts are bound by meCP2 and or PTBP1 before differentiation and not after - that would be consistent with the mechanism presented ... including mRNAs. This could provide a useful resource for other RNAs regulated by the m5C / MeCP2 axis in neuronal differentiation - and likely applicable to many other developmental processes.

I apologize if I have missed m5C any other study that demonstrates such an elegant regulation of RNA. If I have not missed such a study, this truly groundbreaking and could reveal a new layer of RNA regulation that has been missed.

In short, this study will be of immense interest to the general readership of Nature Communications and NPG as a whole. I suggest this study is featured in Nature and other NPG journals.

Reviewer #2 (Remarks to the Author):

In this manuscript, Zhang and coworkers investigate the coordinated role of the noncoding transcript Rncr3 (the host gene of miR124) and miR124 itself in early neurodevelopment. By combining the study of relevant genetic murine models in vivo with mechanistic insights in cells, this study highlights the important roles of these two types of ncRNAs in maintaining the neural progenitor pool and enabling neural differentiation in a timely manner. The findings partially build upon previous work in various labs, which had already demonstrated the role of miR124 in neurodevelopment, while MeCP2 and PTBP1 had previously been reported to bind to the Rncr3 transcript. Nevertheless, one important novelty of this work lies in characterizing the role of exon 2/3 in regulating miR124 biogenesis through MeCP2 recruitment and its binding to PTBP1, which represses the cleavage of the miRNA hairpin. Additionally, the study reveals that MeCP2 binding occurs through m5C-RNA, which had not been described before. A meta-analysis of previous reports indicates that MeCP2 might be a broad reader of m5C-modified transcripts. Overall, the study provides important insights into the role of MeCP2 as an RNA reader and reveals a complex regulatory mechanism orchestrated by ncRNAs with a significant impact on neurodevelopment. However, some aspects of the work are insufficiently supported by the data, and I suggest a revision of some of the experimental approaches.

Major points:

1. In Figure 1 and S1, the authors use CRISPR-mediated deletion of exons 2 and 3 in Rncr3 and ASO-mediated depletion. It is unclear how the different approaches may yield distinct outcomes. While the deletion of the genomic region may still produce a shorter version of Rncr3 containing miR-124, the use of ASOs may completely suppress expression by degrading the entire Rncr3 transcript. This latter case may also impact the production of miR-124. Please provide a more in-depth characterization of the expression of the transcripts from the locus under the different experimental conditions. In order to rule out any unexpected changes in the biogenesis of the lncRNA due to the altered genomic landscape of the locus, sequencing of the shorter Rncr3 RNA needs to be shown in comparison with the full-length transcript upon CRISPR-mediated deletion of exons 2/3 (This may be shown in Fig S2d, but the figure is too small to distinguish whether a new exonic sequence is included in the processed Rncr3 RNA ($\Delta 2/3$). Please clarify.
2. If indeed a new region is included in the $\Delta 2/3$ transcript (see the previous point), experiments should be designed to rule out any positive effect of this region on miR-124a biogenesis.
3. A key contribution of the work is the detailed determinants of MeCP2 binding to Rncr3 RNA. Position 271 within exon 3 is suggested to be the key position that is modified and bound by MeCP2 (Fig 4e, S4e-f), with implied consequences for miR124 regulation. This should be further confirmed by mutating the site and assessing MeCP2 binding, miR124 production, and the associated cellular phenotype. CRISPR-mediated approaches or blocking oligonucleotides are suggested tools to achieve this.
4. In Fig. 4b-g, the results are presented relative to the condition where the maximum signal is obtained. Thus, it is difficult to interpret the data in terms of the percentage of Rncr3 RNA that is indeed m5C methylated on exon 2/3. Quantification of the proportion of Rncr3 that is methylated should be provided to estimate the real impact of the modification on MeCP2 binding and downstream regulation. Along the same lines, bisulfite sequencing in Fig S4e shows that position 271 is highly methylated in MECP2 RIP

experiments, but not in the total population of Rncr3 transcripts (Fig S4d). This seems to indicate that only a low proportion of Rncr3 molecules are bound by MeCP2 in a way driven by the modification of this site.

5. Fig 4 and S4: Given the conservation of c271 and the claim that this is the main modified site recognized by MeCP2, experiments in RenCells should be carried out to test if the same phenomenon occurs in human neural progenitor cells. Is PTBP1-mediated repression of miR124 biogenesis also occurring in human cells? Furthermore, methylation of the RNCR3 transcript, MeCP2 binding to the RNA and PTBP1, and the impact of demethylating agents on protein:RNA interaction and miR124 production should be assessed.

6. Fig 5b,c: How does the binding of DROSHA/DGCR8 to the miR124 hairpin alter upon MeCP2 knockdown?

7. Figure 5d, coIP experiments: It is surprising that the interaction between two abundant proteins such as PTBP1 and MeCP2 is almost completely impaired in the absence of the exon 2/3 sequence. Is the co-IP signal also lost upon NSUN2 (or Trdmt1) knockdown, or Aza treatment?

8. Fig 6: The authors convincingly show that the Lys-rich region within the ID domain is necessary to mediate MeCP2 interaction with Rncr3 RNA. However, additional experiments need to be carried out to show that these residues discriminate between methylated and unmethylated RNA sequences. For example, pulldowns in Fig 6c and 6e should be repeated with an unmethylated oligo, and the decrease in binding compared to that seen for the methylated oligo should be assessed.

Minor Points:

1. Line 183: "The TargetScanMouse website predicted 117 downregulated genes as miR124a targets...". This paragraph is confusing: how many downregulated genes are there in total in the exon2/3 deleted conditions? What are the 101 predicted genes? If the 101 genes show no significant change, why are they labeled in Fig 2d? Are all downregulated genes predicted to be miR-124 targets appearing only in the delta exon2/3 condition, as the graph seems to indicate?

2. Suppl Fig 3d: Quantification of the western blot is needed, as there is uneven loading, as revealed by GAPDH.

3. Fig 4e: Binding to an unrelated oligo sequence should be shown to account for the specificity of the experiment."

Reviewer #3 (Remarks to the Author):

Zhang et al examine lncRNA regulation between Rncr3 and miR124a in mouse embryos and NEPC cell lines using genetic and methylation manipulations. They show Rncr3 regulates NEPC proliferation and is cytosine methylated in exons 2/3. They show MeCP2 binds methyl cytosine RNA in the Exons 2/3 and more generally transcriptome wide. MeCP2 binding prevents miR124a expression and prevents neuronal differentiation consistent with microcephaly in rett syndrome patients. They define the K residues in MeCP2 that act as the methyl cytosine RNA binding domain and recruits PTBP1 to block access by

Drosha/Dgcr8 to miR124a. Overall, the manuscript describes the molecular mechanisms in high detail and uses alternative approaches and cell models as validations that are very thorough and well described. The interpretations are accurate and the findings are novel with respect to the lncRNA regulation but also to MeCP2 as a m5C epitranscriptomic RNA reader protein, and how together they control brain development. Some minor comments are suggested for consideration.

1. ALYREF is raised as a m5C reader with partial homology to the defined MeCP2 K domain and the novelty of this activity in MeCP2 is highlighted in the Discussion. However, apart from the impact on miR124a, could the authors use known roles of ALYREF to speculate on what MeCP2 might do when bound to the global m5C RNA transcriptome?
2. The model in Fig 5e is redundant with parts of Fig 6g. The final box in Fig 5e is not discussed in the text but only in the legend, but is then discussed in the text for Fig 6g. Perhaps Fig 5e could be deleted and just the comprehensive model shown in Fig 6g?
3. As a second approach to examine whether Rncr3 has a small open reading frame that is translated, could the authors examine any published ribosome footprinting data to determine if the transcript is ribosome engaged in the brain?
4. Is it accurate to use $P=0.000$ in some panels (Figs 1h, 2c, 5b, 6b and some supp panels)? In other places it reads <0.0001 which would be a more conventional presentation.
5. Supp Fig 3f shows the increase in NFM at E12.5, but this is not mentioned in the text.

REVIEWER COMMENTS

Reviewer #1 (Remarks to the Author):

The manuscript entitled "m⁵C methylated lncRncr3–MeCP2 interaction restricts miR124a-initiated neurogenesis" describes an interesting new regulation of noncoding RNA via the m⁵C modification and recruitment of MeCP2 therein. This study demonstrates how differential modification can alter RNA biogenesis of LINC00599 (RNCR3), lncRncr3 and miR124a-1 in neurogenesis. The authors show that inhibition of miR124a and genetic knockout of Rncr3 results in decreased NPC proliferation. Intriguingly, homing in on exons 2/3 of Rncr3 results in premature expression of miR124a and in turn, neuronal differentiation processes. The authors further pursue the mechanism of this regulation finding that MeCP2 binds to the exon 2/3 region of Rncr3 and not the miR124a region. Moreover, m⁵C modifications are specified in this same region. This specified site and modifications are differentially regulated during neuronal differentiation pointing to a functional relationship between m⁵C and MeCP2 being fine tuned to initiate neuronal differentiation/proliferation via biogenesis of miR124 and repression of miR124 prior to differentiation. Moreover, the authors find that MeCP2 binding to m⁵C recruits PTBP1 and inhibits miR124a production. Finally the authors identify where this recognition occurs on MePC2.

Together this demonstrates a fascinating new regulatory role of RNA modifications, to my knowledge the first time this has been described. The thorough mechanistic studies reveal even more surprises that DNA methyl binding region of MeCP2 is not required and recognition occurs elsewhere. Thus, there is a fascinating new post-transcriptional regulatory circuit that provides a clock for when miR124 is produced and timing of neuronal differentiation.

I could not find any flaws with the experimentation or interpretations of the data presented. Thus, I do not have any major suggestions or concerns to communicate to the authors.

We are appreciative of the positive statements by reviewer 1 regarding the novelty of our findings and the strength of the experiments and interpretations of the data. We are grateful for their support of our studies.

The only thing I can think to ask is, how general this fascinating mechanism is? The authors could perform RIP or CLIP or fRIP Seq and determine how many other transcripts are bound by meCP2 and or PTBP1 before differentiation and not after - that would be consistent with the mechanism presented ... including mRNAs. This could provide a useful resource for other RNAs regulated by the m⁵C / MeCP2 axis in neuronal differentiation - and likely applicable to many other developmental processes.

We agree with reviewer 1 that the next exciting step will be to determine the generality of the mechanism of MeCP2 binding to other transcripts including m⁵C mRNAs. Our current manuscript is the first demonstration that MeCP2 binds cytosine methylated RNA and we identified an unexplored region and even specific residues of MeCP2 involved in m⁵C binding of Rncr3. In addition, we have done bioinformatic re-analysis of datasets in which the MeCP2 protein binds additional RNAs and which RNAs are cytosine methylated in the brain (**Supplementary Fig. 4k**). The overlap of these two datasets shows that in the mouse brain, ~70% of the m⁵C-modified mRNAs can be simultaneously bound by MeCP2 (**Supplementary Fig. 4k**). **Supplementary Fig. 4l** shows that MeCP2-bound RNAs have significantly higher m⁵C levels than input control at 5x the loading RNA from NE-4C cells. Therefore, we believe these two pieces of evidence support the idea that MeCP2 protein can bind additional methylated RNAs. With these insights and new molecular tools, we are beginning to follow-up on these findings. However, we respectfully feel that this suggested experiment is beyond the scope of the current manuscript and is a separate area of exploration that would take considerable time to conduct and evaluate appropriately.

I apologize if I have missed m⁵C any other study that demonstrates such an elegant regulation of RNA. If I have not missed such a study, this truly groundbreaking and could reveal a new layer of RNA regulation that has been missed.

In short, this study will be of immense interest to the general readership of Nature Communications and NPG as a whole. I suggest this study is featured in Nature and other NPG journals.

We are grateful for the enthusiasm of reviewer 1 and their support of our studies.

Reviewer #2 (Remarks to the Author):

In this manuscript, Zhang and coworkers investigate the coordinated role of the noncoding transcript Rncr3 (the host gene of miR124) and miR124 itself in early neurodevelopment. By combining the study of relevant genetic murine models in vivo with mechanistic insights in cells, this study highlights the important roles of these two types of ncRNAs in maintaining the neural progenitor pool and enabling neural differentiation in a timely manner. The findings partially build upon previous work in various labs, which had already demonstrated the role of miR124 in neurodevelopment, while MeCP2 and PTBP1 had previously been reported to bind to the Rncr3 transcript. Nevertheless, one important novelty of this work lies in characterizing the role of exon 2/3 in regulating miR124 biogenesis through MeCP2 recruitment and its binding to PTBP1, which represses the cleavage of the miRNA hairpin. Additionally, the study reveals that MeCP2 binding occurs through m5C-RNA, which had not been described before. A meta-analysis of previous reports indicates that MeCP2 might be a broad reader of m5C-modified transcripts. Overall, the study provides important insights into the role of MeCP2 as an RNA reader and reveals a complex regulatory mechanism orchestrated by ncRNAs with a significant impact on neurodevelopment. However, some aspects of the work are insufficiently supported by the data, and I suggest a revision of some of the experimental approaches.

We thank Reviewer 2 for their interest in our studies and their support of the novel mechanism of MeCP2 as an RNA reader protein of m5C-modified transcripts.

Major points:

1. In Figure 1 and S1, the authors use CRISPR-mediated deletion of exons 2 and 3 in Rncr3 and ASO-mediated depletion. It is unclear how the different approaches may yield distinct outcomes. While the deletion of the genomic region may still produce a shorter version of Rncr3 containing miR-124, the use of ASOs may completely suppress expression by degrading the entire Rncr3 transcript. This latter case may also impact the production of miR-124. Please provide a more in-depth characterization of the expression of the transcripts from the locus under the different experimental conditions. In order to rule out any unexpected changes in the biogenesis of the lncRNA due to the altered genomic landscape of the locus, sequencing of the shorter Rncr3 RNA needs to be shown in comparison with the full-length transcript upon CRISPR-mediated deletion of exons 2/3 (This may be shown in Fig S2d, but the figure is too small to distinguish whether a new exonic sequence is included in the processed Rncr3 RNA (delta2/3). Please clarify.

ASO was done under proliferation conditions when miR124a levels are very low. To more firmly establish this point, we now moved old Fig 2a up to become new Fig 1c. This shows the strong expression of Rncr3 (exon 2/3 probe) in neuroepithelial cells /neural progenitors but no detectable expression of miR124a until the onset of differentiation (PCR also confirms very low miR124a expression in proliferating NEPCs). ASO treatment was indeed done to suppress expression by degrading the entire Rncr3 transcript. This small aspect of the paper was a prelude to provide preliminary evidence that Rncr3 has a role in neuroprogenitor cells. We immediately followed this up by CRISPR gene editing of either the entire Rncr3 locus or more specific deletion of exons 2/3 (leaving miR124a intact). The ASO and gene editing results are consistent showing an important regulatory role for Rncr3 in proliferation of stem cells and protection of neural stem cells from apoptosis.

In terms of the genomic locus, we used CRISPR/Cas9 gene editing to create a null cell line in which the entire Rncr3 locus was deleted and another cell line with local deletion of the regions of exons 2 and 3. In the exon 2&3 deletion line, sequencing of the genomic locus shows a clean deletion but the resulting RNA transcript includes a cryptic exonic fragment between exons 1 & 4 (enlarged sequence in **Supplementary Fig. 2c** and **2d**). Please see the point below in which we created a clean exons 2&3 deletion construct to ensure that the cryptic exon does not contribute to functionality.

2. If indeed a new region is included in the Delta2/3 transcript (see the previous point), experiments should be designed to rule out any positive effect of this region on miR-124a biogenesis.

We are grateful to the reviewer for this suggestion. We have carefully considered this suggestion and carried out the experimental design. New data to clarify any negative or positive effects of this cryptic exon region on miR-124a biogenesis is shown in **Supplementary Fig. 2c-f**. Here we established different constructs to express wildtype Rncr3, Rncr3 exon2/3 deletion with the cryptic exon (Δ exon2&3+) or without the cryptic exon region (Δ exon2&3) and transfected the different plasmids into Rncr3 full length KO cells to assay the level of miR-124a production. The results (**Supplementary Fig. 2f**) show full-length wild type Rncr3 (containing exons 2&3) shows limited miR-124a expression (17.5 fold over background in the KO); while the two plasmids of Δ exon2&3 and Δ exon2&3+ induced miR124a by 91.7-fold and 87.7-fold, respectively. Importantly, this shows no significant difference between Δ exon2&3 and Δ exon2&3+ constructs in terms of miR124a biogenesis.

3. A key contribution of the work is the detailed determinants of MeCP2 binding to Rncr3 RNA. Position 271 within exon 3 is suggested to be the key position that is modified and bound by MeCP2 (Fig 4e, S4e-f), with implied consequences for miR124 regulation. This should be further confirmed by mutating the site and assessing MeCP2 binding, miR124 production, and the associated cellular phenotype. CRISPR-mediated approaches or blocking oligonucleotides are suggested tools to achieve this.

We agree with this comment from the reviewer and designed experiments to address this question. First, we created three Rncr3 plasmids, which express mRNAs of wild-type Rncr3, or 271C>T, or 271-313 C>Ts (including 271/272/274/280/297/300/302/313 cytosine residues). For each plasmid, the 3' end of the Rncr3 mRNA contains the MS2 binding hairpins recognized by the fusion protein of MS2-GFP, which allows for mRNA-protein complex purification using anti-MS2 antibody-conjugated beads. In this way we can enrich for cross-linked Rncr3-protein complex by MS2 antibody and then use western blotting to quantify MeCP2 binding with the different genotypes of Rncr3. **Supplementary Fig. 4i** shows that indeed the binding of MeCP2 is significantly decreased by mutation of 271C>T and 271-313 C>Ts of Rncr3. Moreover, **Supplementary Fig. 4m** shows that the level of mature miR124a is significantly increased by mutations of 271C>T and 271-313 C>Ts.

4. In Fig. 4b-g, the results are presented relative to the condition where the maximum signal is obtained. Thus, it is difficult to interpret the data in terms of the percentage of Rncr3 RNA that is indeed m5C methylated on exon 2/3. Quantification of the proportion of Rncr3 that is methylated should be provided to estimate the real impact of the modification on MeCP2 binding and downstream regulation. Along the same lines, bisulfite sequencing in Fig S4e shows that position 271 is highly methylated in MECP2 RIP experiments, but not in the total population of Rncr3 transcripts (Fig S4d). This seems to indicate that only a low proportion of Rncr3 molecules are bound by MeCP2 in a way driven by the modification of this site.

Thanks to the reviewer for their appreciation of the dynamic regulation of Rncr3 methylation-MeCP2 binding-miR124a expression that our study demonstrates. We do recognize that previous **Supplementary Fig. 4 g** bisulfite sequencing results show variability in Rncr3 methylation pattern with single m5C modification, modification at multiple sites and variability in the extent of C271 modification. In part this is likely due to **1**) limitation in the methodology in which we

sequenced a limited number of bisulfite treated RNA-RT PCR clones which cannot fully represent the m5C modifications in this region in cells. At the same time, this is why we screened a total of 50 bisulfite treated RNA-RT PCR clones in **Supplementary Fig. 4g** to confirm the complete disappearance of m5C modification in this region in cells after neural differentiation. **2)** cytosine residues at positions 271-313 within Rncr3 exon 3 are conserved in humans. Therefore, we believe that the regulation of the binding interaction between MeCP2 protein and Rncr3 molecules in this region comes from the contribution of multiple methylated cytosines. **3)** this variability likely reflects the biology of an *in vivo* system - we are not aware of sequence specificity of RNA methyltransferases and the placing of m5C modifications may well depend on other aspects of RNA biology such as structure and presence or absence of other proteins on the RNA. **4)** as shown in **Supplementary Fig. 4i**, even if the cytosine residues at multiple positions of C271 or C271-313 are mutated, Rncr3 can still show weak binding to MeCP2, in fact even the oligo probe without C271 methylation modification shows weak binding to MeCP2 (**Supplementary Fig. 4h**). Therefore, we suspect that additional mechanisms contribute to MeCP2 binding of Rncr3. As pointed out in the original discussion, “We speculate that the RG-repeat RNA binding modules in the ID [intervening domain of MeCP2] provides affinity for RNA (currently thought to have higher affinity for G-quadruplex or GC-rich dsRNA)^{44,65} and that the lysine-rich region specifies binding to cytosine methylated RNA.”

Our on-going studies that are outside of the scope of this current manuscript are designed to extend the analysis of MeCP2 binding to Rncr3 using differentially methylated Rncr3 oligos (single sites, multiple methylated sites) and different RNA structures (hairpin, dsRNA, etc.) to more precisely determine how MeCP2 interacts with m5C and unmodified RNAs.

5. Fig 4 and S4: Given the conservation of c271 and the claim that this is the main modified site recognized by MeCP2, experiments in RenCells should be carried out to test if the same phenomenon occurs in human neural progenitor cells. Is PTBP1-mediated repression of miR124 biogenesis also occurring in human cells? Furthermore, methylation of the RNCR3 transcript, MeCP2 binding to the RNA and PTBP1, and the impact of demethylating agents on protein:RNA interaction and miR124 production should be assessed.

Many thanks to the reviewer for posing an interesting extension of our studies. Relative to miR124 and PTBP1, most published papers focused on the regulatory effect of miR124 on PTBP1. Only one study by Yeom et al. in 2018 clearly showed the inhibitory effect of PTBP1 on miR124 in mouse ESCs. They showed that PTBP1 directly binds to a CU-rich segment in pri-miR-124-1 (RNCR3), thereby blocking DROSHA and DGCR8 on pri-miR-124-1, to inhibit the processing of miR124. This regulatory mechanism highlighted a “CU-rich segment” of 107nt in the 5' flank of the miR-124 stem-loop sequence in pri-miR-124-1. Our analysis in response to this comment shows a partially conserved CU-rich segment in human *RNCR3* (79nt, conserved identity 73%). Notably, a completely conserved PTBP1 binding motif "CCUCUCUCUC" is closely adjacent to the 5' of miR-124 stem-loop in mouse and human pri-*miR-124-1*. This suggests that PTBP1 protein is likely to directly bind to pri-miR-124-1 in human cells and exert a regulatory function. Furthermore, all cytosine residues at positions 271-313 within Rncr3 exon 3 are conserved in humans. As discussed above, methylated cytosine 271 within Rncr3 exon 3 has a strong affinity for the MeCP2 protein, but multiple cytosine residues in this region are methylated and may all contribute to the binding of MeCP2 protein.

In this revision, we include new experiments in human neuroprogenitor cells. **New Fig. 4d** shows m5C RIP data indicating cytosine methylation modifications in the conserved region of human RNCR3; these modifications are increased by folic acid treatment; and 5AZA treatment abolishes this signal, whereas **supplementary Fig. 4c** non-conserved region of RNCR3 does not show significant m5C methylation. **New Fig. 4d** shows RNCR3 conserved region is bound by MeCP2, and this binding is disrupted by 5AZA treatment. In addition, **new Fig. 5b** using co-IP shows that the binding between MeCP2 and PTBP1 is increased by folic acid treatment and significantly decreased by 5AZA treatment. Correspondingly, **new Fig. 4j** shows significantly increased processing of miR124a following 5AZA treatment. Thus, *the mechanisms that we have defined hold true for both mouse and human neural progenitor cells whereby conserved exon 3 of Rncr3/RNCR3 are cytosine methylated, bound by MeCP2, which serves to recruit PTBP1 to a small conserved motif upstream of the miR-124 stem-loop, to prevent DROSHA/DGCR8 (see point 6 below) and repress miR124a biogenesis, serving to maintain neural stem cell proliferation.*

6. Fig 5b,c: How does the binding of DROSHA/DGCR8 to the miR124 hairpin alter upon MeCP2 knockdown?

New Fig. 5f and 5g show that after MeCP2 knockdown, the binding of DROSHA and DGCR8 to the miR-124 stem-loop region in Rncr3 is increased.

7. Figure 5d, coIP experiments: It is surprising that the interaction between two abundant proteins such as PTBP1 and MeCP2 is almost completely impaired in the absence of the exon 2/3 sequence. Is the co-IP signal also lost upon NSUN2 (or Trdmt1) knockdown, or Aza treatment?

Indeed, as mentioned above, 5AZA treatment significantly reduced the binding between MeCP2 protein and PTBP1 protein in human neural progenitor cells (**Fig. 5b**).

8. Fig 6: The authors convincingly show that the Lys-rich region within the ID domain is necessary to mediate MeCP2 interaction with Rncr3 RNA. However, additional experiments need to be carried out to show that these residues discriminate between methylated and unmethylated RNA sequences. For example, pulldowns in Fig 6c and 6e should be repeated with an unmethylated oligo, and the decrease in binding compared to that seen for the methylated oligo should be assessed.

We believe this has already been adequately shown through a combination of Supplementary Fig 4h and Fig 6e. Supplementary Fig. 4h showed that wildtype MeCP2 binds more strongly to the methylated C271 oligo than the unmethylated oligo. By comparison, the binding efficiency of wild-type MeCP2 protein to the un-methylated probe (Supplementary Fig 4h) is basically same as the binding efficiency of the lysine-mutated MeCP2 to the methylated probe (Fig. 6c and e). Again, we do not mean to give the impression that Rncr3 m5C 271 residue is the sole mediator for binding to MeCP2 as some binding occurs even to the unmethylated oligo. We have de-emphasized residue 271 in the revised manuscript. Nonetheless, our results clearly show the importance of cytosine methylation for binding between MeCP2 and Rncr3, and that the MeCP2 lysine sites are also indispensable for binding to methylated Rncr3.

Minor Points:

1. Line 183: "The TargetScanMouse website predicted 117 downregulated genes as miR124a targets...". This paragraph is confusing: how many downregulated genes are there in total in the exon2/3 deleted conditions? What are the 101 predicted genes? If the 101 genes show no significant change, why are they labeled in Fig 2d? Are all downregulated genes predicted to be miR-124 targets appearing only in the delta exon2/3 condition, as the graph seems to indicate?

We apologize for lack of clarity. Below we answer the relevant questions raised by the reviewers one by one. We have added this information to the Methods section.

1) how many downregulated genes are there in total in the exon2/3 deleted conditions?

In exons 2/3 deleted cells, there were a total of 878 genes whose expression was significantly reduced compared with wild type, and the overall change range (log2FoldChange) is from -0.52 to -9.52.

2) What are the 101 predicted genes?

Since our experimental evidence has shown that exon2/3 deletion leads to a significant up-regulation of the levels of mature miR124a, we speculated that the levels of mRNAs that are regulated by miR124a would show a downward trend. Therefore, we created a list of all potential target genes of miR124a (a total of 1580 genes) through the analysis of the TargetScanMouse website, and then compared this miR124a target list with the list of 878 down-regulated genes in exon2/3 deletion cells. The number of overlapping genes between the two lists was 117. We further considered that among these 117 down-regulated genes, genes that are also down-regulated in Rncr3KO cells, would not be miR124a-regulated target genes. Among the 117 down-regulated genes, 16 genes were down-regulated in both exon2/3 deletion cells and Rncr3 KO cells, and these were excluded. The remaining 101 miR124a target genes that were only down-regulated in exon2/3 deletion cells and

remained unchanged in Rncr3 KO cells were selected and shown in Fig. 2e, and the specific gene information is shown in Source data of table Fig 2e.

3) If the 101 genes show no significant change, why are they labeled in Fig 2d?

Please see the answer above. These 101 genes were down-regulated in exon2/3 deletion cells, **but remained unchanged in Rncr3 KO cells.**

4) Are all downregulated genes predicted to be miR-124 targets appearing only in the delta exon2/3 condition, as the graph seems to indicate?

Yes, these 101 genes are all downregulated genes predicted to be miR124 targets appearing only in the delta exon2/3 cells, supporting the earlier information that miR124 levels are increased in delta exon2/3 cell line.

2. Suppl Fig 3d: Quantification of the western blot is needed, as there is uneven loading, as revealed by GAPDH.

Quantitative analysis data has been added to the revised manuscript.

3. Fig 4e: Binding to an unrelated oligo sequence should be shown to account for the specificity of the experiment.

In order to verify the specificity of the ability of MeCP2 protein to recognize the methylation modification site me271C in Rncr3, as shown in Supplementary Fig. 4h, we created an oligo with sequences surrounding Rncr3 residue 323 as a negative control, and artificially added m5C modification on 323C to test for MeCP2 binding. The results showed that the artificially added m5C modification did not enhance MeCP2 binding.

Reviewer #3 (Remarks to the Author):

Zhang et al examine lncRNA regulation between Rncr3 and miR124a in mouse embryos and NEPC cell lines using genetic and methylation manipulations. They show Rncr3 regulates NEPC proliferation and is cytosine methylated in exons 2/3. They show MeCP2 binds methyl cytosine RNA in the Exons 2/3 and more generally transcriptome wide. MeCP2 binding prevents miR124a expression and prevents neuronal differentiation consistent with microcephaly in rett syndrome patients. They define the K residues in MeCP2 that act as the methyl cytosine RNA binding domain and recruits PTBP1 to block access by Drosha/Dgcr8 to miR124a. Overall, the manuscript describes the molecular mechanisms in high detail and uses alternative approaches and cell models as validations that are very thorough and well described. The interpretations are accurate and the findings are novel with respect to the lncRNA regulation but also to MeCP2 as a m5C epitranscriptomic RNA reader protein, and how together they control brain development. Some minor comments are suggested for consideration.

1. ALYREF is raised as a m5C reader with partial homology to the defined MeCP2 K domain and the novelty of this activity in MeCP2 is highlighted in the Discussion. However, apart from the impact on miR124a, could the authors use known roles of ALYREF to speculate on what MeCP2 might do when bound to the global m5C RNA transcriptome?

This is an interesting question but as of now the information is too sparse and premature to speculate in the manuscript. However, here are our thoughts for this forward-looking question: the roles of MeCP2 bound to the global m5C RNA transcriptome could be speculated as 1) assist in alternative splicing in specific cell types, 2) help in the recruitment of specific cytosine methylated mRNAs to partner proteins to form protein-RNA complexes; or 3) in the formation of liquid phase separations which are closely related to different cellular physiological processes or pathological processes.

2. The model in Fig 5e is redundant with parts of Fig 6g. The final box in Fig 5e is not discussed in the text but only in the legend, but is then discussed in the text for Fig 6g. Perhaps Fig 5e could be deleted and just the comprehensive model shown in Fig 6g?

We have deleted figure 5 model in the revised manuscript.

3. As a second approach to examine whether Rncr3 has a small open reading frame that is translated, could the authors examine any published ribosome footprinting data to determine if the transcript is ribosome engaged in the brain?

We thank the reviewer for this helpful suggestion. We followed the reviewer's suggestion and searched the ribosome footprinting RNA-seq database of published literature (GWIPS-viz; <https://gwips.ucc.ie/>; Ingolia et al (2009) Science; doi: 10.1126/science.1168978.). We first examined the housekeeping gene, *Actb*; the nervous system-specific expression gene *Sox2*; and the classic lncRNA, *Xist*. These were used as controls for analysis. For the coding genes *Actb* and *Sox2*, there is strong ribosome footprinting data across these genes but for lncRNA *Xist* there is minimal footprinting across the gene (see 1st three figures below). We then examined the data of *Rncr3* (4th figure below) and found similar and minimal distribution characteristics as seen for *Xist*. These data provide additional evidence that *Rncr3* does not have coding potential.

Actb gene

Sox2 gene

Xist gene

Rncr3 gene

4. Is it accurate to use $P=0.000$ in some panels (Figs 1h, 2c, 5b, 6b and some supp panels)? In other places it reads <0.0001 which would be a more conventional presentation.

We have corrected this in the revised manuscript.

5. Supp Fig 3f shows the increase in NFM at E12.5, but this is not mentioned in the text.

We now mention this in the revised manuscript.

REVIEWERS' COMMENTS

Reviewer #1 (Remarks to the Author):

My minor concerns have been addressed - thank you and congratulations to the authors on an impressive, important and insightful study!

Reviewer #2 (Remarks to the Author):

The authors have satisfactorily addressed my comments, specifically by providing new experiments to clarify some important issues:

- They have ensured that the inclusion of cryptic sequences in the edited Rncr3 gene does not influence the results.
- They have confirmed the site of interaction between MeCP2 and Rncr3.
- They have carried out experiments that demonstrate the conservation of the regulatory mechanism in humans.

I am satisfied with the explanations and new data provided, and I consider the work suitable for publication

Reviewer #3 (Remarks to the Author):

All concerns raised have been effectively addressed.